



# Global Atmospheric Chemistry – Which Air Matters

Michael J. Prather[1], Xin Zhu[1], Clare M. Flynn[1], Sarah A. Strode[2,3], Jose M. Rodriguez[3], Stephen

D. Steenrod[2,3], Junhua Liu[2,3], Jean-Francois Lamarque[4], Arlene M. Fiore[5], Larry W. Horowitz [6],

Jingqiu Mao[7], Lee T. Murray[8], Drew T. Shindell[9], Steven C. Wofsy[10]

[1]Department of Earth System Science, University of California, Irvine, CA 92697-3100, USA

[2]NASA Goddard Space Flight Center, Greenbelt, MD, USA

[3]Universities Space Research Association (USRA), GESTAR, Columbia, MD, USA

[4]Atmospheric Chemistry, Observations & Modelling Laboratory, National Center for Atmospheric Research,
      Boulder, CO 80301, USA

[5]Department of Earth and Environmental Sciences and Lamont-Doherty Earth Observatory of Columbia University,

Palisades, New York, USA

[6]Geophysical Fluid Dynamics Laboratory, National Oceanic and Atmospheric Administration, Princeton, NJ, USA

[7]Geophysical Institute and Department of Chemistry, University of Alaska Fairbanks, Fairbanks, Alaska, USA

[8]Department of Earth and Environmental Sciences, University of Rochester, Rochester, NY 14627-0221 USA

[9]Nicholas School of the Environment, Duke University, Durham, NC, USA

[10]School of Engineering and Applied Sciences, Harvard University, Cambridge, Massachusetts, 02138 USA



***Abstract.*** An approach for analysis and modeling of global atmospheric chemistry is developed for application to measurements that provide a tropospheric climatology of those heterogeneously distributed, reactive species that control the loss of methane and the production and loss of ozone. We identify key species (e.g., $O_3$, $NO_x$, $HNO_3$, $HNO_4$, $C_2H_3NO_5$, $H_2O$, HOOH, $CH_3OOH$, HCHO, CO, $CH_4$, $C_2H_6$, acetaldehyde, acetone), and presume that they can be measured simultaneously in air parcels on the scale of a few km horizontally and a few tenths vertically. Six global models have prepared such climatologies (at model resolution) for August with emphasis on the vast central Pacific and Atlantic Ocean basins. We show clear differences in model generated reactivities as well as species covariances that could readily be discriminated with an unbiased climatology. A primary tool is comparison of multi-dimensional probability densities of key species weighted by frequency of occurrence as well as by the reactivity of the parcels with respect to methane and ozone. The reactivity-weighted probabilities tell us which parcels matter in this case. Testing 100-km scale models with 2-km measurements using these tools also addresses a core question about model resolution and whether fine-scale atmospheric structures matter to the overall ozone and methane budget. A new method enabling these six global chemistry-climate models to ingest an externally-sourced climatology and then compute air parcel reactivity is demonstrated. Such an observed climatology is anticipated from the NASA Atmospheric Tomography (ATom) aircraft mission (2015-2020), measuring the key species, executing profiles over the Pacific and Atlantic Ocean basins. This work is a core part of the design of ATom.



## *1. Introduction*

To understand global atmospheric chemistry is to understand the chemical heterogeneity of air
parcels across the vastness of the troposphere (e.g., Fishman et al., 1996; Ehhalt et al., 1997;
Marenco et al., 1998; Jacob et al., 2003; Olson et al., 2004; Kunz et al., 2008; Jacob et al., 2010;
Nicely et al., 2016). These air parcels are ephemeral, being continually created, evolving, and
mixed with others. Even the concept of discrete as opposed to a continuum of air parcels is a

conceit based in part on our modeling of the atmosphere in quantized units such as gridded cells
or 1-second averages. Yet, the concept of distinct air parcels remains useful for parsing in situ
aircraft measurements and for the analysis presented here in which we ask which air is more
important for the chemical evolution of human-driven air pollution.

To understand the mix of chemicals in the atmosphere is to recognize how humans have
perturbed the common air we breathe. We seek knowledge of the photochemical evolution in
each air parcel to understand the overall impact of this heterogeneity and to interpret past
changes and predict future ones. The integration of this chemical reactivity over the ensemble of
such heterogeneous air masses controls atmospheric residence times of air pollutants and reactive

greenhouse gases, particularly methane and ozone. Hence, it allows us to evaluate the
consequences of many atmospheric pollutants as regards global air quality and climate.

We have a tendency to simplify this heterogeneity as global, hemispheric, or even regional
averages that can be represented with an average chemical composition. This holds true

especially when diagnosing the sources and sinks of critical pollutants, or when comparing
models with atmospheric measurements. Yet, chemistry inherently involves quadratic reactions
of two or more species and hence is non-linear – viz. the chemistry integrated over a mix of
parcels is not necessarily the same as that over the average of the mix. We have progressed in
modeling atmospheric chemistry over the past four decades from a few boxes (e.g., stratosphere

and troposphere, northern and southern hemispheres) to high-resolution gridded models with
many millions of cells. These models simulate myriads of air parcels that at times represent the
observed atmospheric heterogeneity. For example, Figure 1a presents a single-day snapshot of
the column loss of methane as simulated by the UC Irvine chemistry-transport model (CTM) at a





resolution of 1 degree in latitude and longitude. Even column averages over 24 hours show a

filamentary structure with most of the tropospheric loss of methane occurring in sharp synoptic

patterns. These chemical patterns have similarities with the atmospheric rivers of water vapor

(Newell et al., 1992; Dacre et al., 2015; Mundhenk et al., 2016), and both patterns are dominated

by the lower half of the troposphere. Nevertheless, the methane-loss filaments do not coincide

with atmospheric rivers (Figure 1a vs. 1b), indicating that chemical heterogeneity plays a role in

these fine-scale structures, (e.g., Ehhalt et al., 1997; Browell et al., 2003; Charlton-Perez et al.,

2009).

This heterogeneity of species and chemical reactivity (e.g., methane loss) is clearly structured

and not simply Gaussian. Its structure reflects the combined influence of meteorological

transport and mixing as well as the ways that different species are co-emitted and transformed

around the globe. For example, large plumes from industrial regions or biomass burning when

lofted into the free troposphere by deep convection or frontal systems will naturally be sheared

into laminae, travel long distances, and appear ubiquitously (Newell et al., 1999; Stoller et al.,

1999; Singh et al., 2000; Blake et al., 2003; Heald et al., 2003; Damoah et al., 2004; Hecobian et

al., 2011; Wofsy et al., 2011). This shear or random strain in the atmosphere acts to maintain the

pollution concentrated within the layer and preserves the sharp gradients relative to the

neighboring atmosphere before they dissolve into the surrounding atmosphere, e.g., (Prather and

Jaffe, 1990; Thuburn and Tan, 1997; Esler, 2003; Pisso et al., 2009). Characterizing chemical

species in the atmosphere as having mean abundances, or even mean vertical profiles, with a

standard deviation to represent the observed variability, does not really describe how these

models generate heterogeneity and how the different species co-vary. Assuredly, the atmosphere

has more processes and structures than are in our current, high-resolution models as seen in

Figure 1, but the extent to which these models represent the key processes shaping the observed

patterns is understudied.


Characterizing atmospheric measurements of this chemical heterogeneity specifically for testing

models is problematic. Simple direct comparisons of atmospheric rivers, pollution or biomass

burning plumes, and other structures in the troposphere or stratosphere is difficult, even with

models using the historical meteorology and chemical emissions, because of slight phase errors



in the location of large-scale gradients or laminae (e.g., Reid et al., 1998; Manney et al., 1998; Wild et al., 2003; Kiley et al., 2003; Allen et al., 2004; Schoeberl et al., 2007; Elguindi et al., 2010). The other type of chemistry models, the chemistry-climate models (CCMs), are our means of understanding future air pollution (Prather et al., 2003; Mickley et al., 2004; Jacob and Winner, 2009; Fiore et al., 2012; Barnes and Fiore, 2013; Turner et al., 2013; Fang et al., 2013;

Schnell et al., 2015), but CCMs describe the chemical climate and not the hindcast of specific chemical measurements. Most large CCM groups have a parallel CTM versions, but these forced-meteorology versions will likely have different clouds, convection and transport, changing the chemical climatology. Aircraft campaigns often use a photochemical box models to provide an observationally constrained check on reactive species (Olson et al., 2004; Apel et

al., 2012; Olson et al., 2012; Stone et al., 2012), and more recently these have extended the box model as a transfer standard across CCM/CTMs (Nicely et al., 2016) that can integrate reactive chemistry over 24 hours. The problem remains that the 24-hour integration requires a global model's diagnostics for the diurnal cycle of cloud cover and ozone/aerosol influence on photolysis.


We describe a new approach for developing chemical climatologies from atmospheric chemistry measurements, and for using the major global 3D CTM/CCMs as box-models to integrate the 24-hour rates for important species like methane and ozone. Our goal is to provide climatologies that can point to specific patterns of the key chemical species whose initial values control the

chemical evolution of the air parcels. Knowing the correct multi-species patterns, and how different models succeed or fail in reproducing them, will give developers the largest leverage in improving the chemical and physical processes in the models. A critical issue in preparing such a chemical climatology is representativeness, i.e., just how well do the observations represent the region in which they were made and how well should the models match the space-time

frequency of the observations. There is a growing literature on the issues of representativeness of atmospheric measurements (Nappo et al., 1982; Crawford et al., 2003; Hsu et al., 2004; Ramsey and Hewitt, 2005; Larsen et al., 2014; Eckstein et al., 2016) including defining the chemical patterns through cluster analysis (Koppe et al., 2009). Most atmospheric chemistry missions have had a specific focus (Jacob et al., 2003; Engel et al., 2006; Jacob et al., 2010; Pan

et al., 2016) and thus produce a biased, non-climatological sampling, for example, by chasing



pollution plumes (Hsu et al., 2004) or by measuring only in clear skies (Nicely et al., 2016). We examine below some aspects of making objective climatologies of chemical observations, in particular the representativeness of atmospheric transects over the remote ocean basins. Our approach was designed specifically as part of the current NASA Atmospheric Tomography

(ATom) aircraft mission in which the DC-8 is instrumented to make high-frequency in situ measurements of the most important reactive species and flies down the middle of the Pacific and Atlantic Oceans, profiling as frequently as possible. The resulting climatology represents the heterogeneity of the atmosphere, including the co-variance of key reactive species.

This approach is tested here using six CTM/CCMs described in Table 1. The model versions are used here as examples. They are meant to demonstrate the methodology and the ability to discriminate among them with ATom-like measurements. No model tuning or development occurred as part of this work, except to correct where quantities were missed or misdiagnosed. These diagnostics will need to be revisited for the model versions used in assessments (e.g.,

Lamarque et al., 2013; Collins et al. 2016)

Typically, the probability of occurrence of a species' abundance is weighted by the air mass of the parcel, but if we are interested in the chemical reactivity, then the parcel should be weighted by the chemical rates in the parcel (e.g., moles per day). Such weighting is an obvious choice in

that it tells us which air parcels matter for chemical budgets, including, for example, whether infrequently observed pollution plumes are responsible for a large fraction of ozone production.

In Section 2 we define our use of reactivity in this paper (i.e., the production and loss of ozone, the loss of methane) and identify about a dozen key chemical species and other variables that

once initialized determine the chemical evolution of an air parcel. In Section 3 we show how the CTM/CCMs can be altered slightly to calculate the reactivity of air parcels using the native grid cells of the model and a prescribed initialization of the key chemical species. This approach allows the CTM/CCMs to be run using either model data or observations, or a mixture of both. In Section 4 we derive multi-dimension probability distributions for these key variables over a

suitable latitude-longitude-pressure domain using grid-cell values from several CTM/CCMs. These distributions clearly show the basic differences in chemical heterogeneity across the six



models. We conclude in Section 5 with a discussion of the ongoing NASA ATom mission (2015-2020), which will provide the air parcel measurements of key species to initialize the models' calculation of reactivity in each parcel and thus provide an observed climatology of the

chemical reactivity of the troposphere. This approach moves us towards an understanding of which species exert the largest influence on the atmosphere, and thus which are thus most crucial for us to establish a global climatology.



| Table 1a. Participating models | | | | | |
|---|---|---|---|---|---|
| model | type | driving meteorology | year | model grid | effective resol @ 500hPa |
| CAM4-Chem | CCM | SSTs | 2000s | 0.47°x0.625°x52L | 0.47° x 0.625°x 38hPa |
| GEOS-Chem | CTM | GEOS5-FP | 2013 | 2° x 2.5° x 72L | 2° x 2.5° x 38hPa |
| GFDL-AM3 | CCM | NCEP (nudged) | 2013 | C180L48 | 0.5° x 0.5° x 71hPa |
| GISS-E2 | CCM | Daily SSTs prescribed, winds nudged to MERRA | 2013 | 2° x 2.5° x 40L | 2° x 2.5° x 50hPa |
| GMI-CTM | CTM | MERRA | 2001 | 1° x 1.25° x 72L | 1° x 1.25° x 38hPa |
| UCI-CTM | CTM | ECMWF IFS Cy38r1 | 2005 | T159N80L60 | 1.1° x 1.1° x 38hPa |

| Table 1b. | | | |
|---|---|---|---|
| model | POC | email | model url |
| CAM4-Chem | Jean-Francois Lamarque | lamar@ucar.edu | http://www.cesm.ucar.edu/models/current.html |
| GEOS-Chem | Lee Murray | lee.murray@rochester.edu | http://geos-chem.org__ver 10-01 |
| GFDL-AM3 | Arlene Fiore | amfiore@ldeo.columbia.edu | https://www.gfdl.noaa.gov/am3-model/ |
| GISS-E2 | Lee Murray | lee.murray@rochester.edu | http://www.giss.nasa.gov/tools/modelE/ |
| GMI-CTM | Sarah Strode | Sarah.A.Strode@nasa.gov | http://gmi.gsfc.nasa.gov |
| UCI-CTM | Michael Prather | mprather@uci.edu | ftp://halo.ess.uci.edu/public/xzhu/qcode_72c |


| Table 1c. | | |
|---|---|---|
| model | graph label | relevant refs |
| CAM4-Chem | A | (Lamarque et al., 2012;Tilmes et al., 2016) |
| GEOS-Chem | B | (Bey et al., 2001;Eastham et al., 2014) |
| GFDL-AM3 | C | (Donner et al., 2011;Naik et al., 2013a;Li et al., 2016) |
| GISS-E2 | D | (Schmidt et al., 2014;Shindell et al., 2013) |
| GMI-CTM | E | (Strahan et al., 2007;Duncan et al., 2007) |
| UCI-CTM | F | (Holmes et al., 2013;Holmes et al., 2014;Prather, 2015;Sovde et al., 2012) |

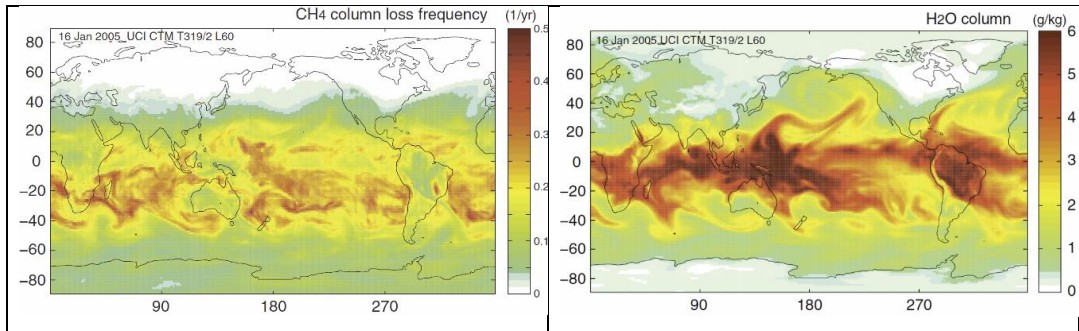

**Figure 1.** (a) Column tropospheric loss frequency (1/yr) for $CH_4$ and (b) column average $H_2O$ abundance (g-$H_2O$/kg-air) taken from a 1-day integration (16 Jan 2005) using the University of California Irvine (UCI) chemistry-transport model (CTM) run at T319N80L57 resolution (~1° horizontal) using forecast meteorology from the European Centre for Medium-Range Weather Forecasts, see Sovde et al. (2012). As expected, the northern winter shows very little $CH_4$ loss above 40N.



## 2. Key chemical species for tropospheric reactivity

The reactivity of an air parcel is defined here as a daily average of the rates affecting critical
species, in this case, ozone ($O_3$), a greenhouse gas and air quality threat, and methane ($CH_4$), the
second most important emitted greenhouse gas after $CO_2$. Methane is emitted mostly through
human activities but also naturally; and it is lost primarily (>80%) through reaction with the
hydroxyl radical (OH) in the troposphere (reaction 1). Other atmospheric losses in decreasing
order of magnitude and certainty are reaction with stratospheric OH, surface uptake by biota, and
reaction with Cl atoms (Prather et al., 2012; Ciais et al., 2013).

$$CH_4 + OH \rightarrow CH_3 + H_2O \tag{1}$$

The $CH_4$ abundance varies little throughout the troposphere (~10%), and the destruction of $CH_4$
occurs with a mean loss frequency of ~0.1 /yr (see Fig. 1a). Here we focus on calculating the
tropospheric loss of $CH_4$ by OH over 24 hours (reaction 1, designated L-CH4) in units of ppb
(nanomoles/mole-air) per day.

Tropospheric $O_3$ has stratospheric sources and surface sinks, which average to about 0.2 - 0.3
ppb per day, and much larger in situ photochemical production and losses that average about 1.1
- 1.5 ppb per day (Stevenson et al., 2006; Stevenson et al., 2013; Young et al., 2013; Hardacre et
al., 2015). The $O_3$ abundance varies greatly throughout the troposphere, by a factor of 10 or
more, and its mean residence time is about a month (Stevenson et al., 2006; Wu et al., 2007; Hsu
and Prather, 2009). $O_3$ is an intermediate source of atomic O in many tropospheric reactions,
and its net production and loss is determined in the long term by the breaking and reforming of
the O-O bond originating with molecular oxygen. Chemical reactions are traditionally grouped
into production (P-O3, ppb/day)

$$HO_2 + NO \rightarrow NO_2 + OH \tag{2a}$$
$$RO_2 + NO \rightarrow NO_2 + RO \tag{2b}$$
$$\text{where} \quad NO_2 + hv \rightarrow NO + O \quad \text{and} \quad O + O_2 \rightarrow O_3 \tag{2c}$$
$$O_2 + hv \rightarrow O + O \quad \text{(times 2)} \tag{2d}$$

and loss (L-O3, ppb/day).

$$O_3 + OH \rightarrow O_2 + HO_2 \tag{3a}$$
$$O_3 + HO_2 \rightarrow HO + O_2 + O_2 \tag{3b}$$



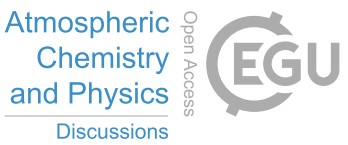

$$O(^1D) + H_2O \rightarrow OH + OH \hspace{3cm} (3c)$$

$\hspace{1.5cm}$ where $O_3 + h\nu \rightarrow O(^1D) + O_2$ $\hspace{3cm}$ (3d)

In the troposphere, reaction 2d is important only in the tropics above 12 km (Prather, 2009). The true P minus L of $O_3$ includes a large number of other reactions, particularly involving oxides of nitrogen and hydrocarbons; but throughout the remote troposphere (i.e., away from fresh pollution sources), reactions (2) minus reactions (3) accurately approximate the true P – L that

the models calculate using the full set of reactions. One reason for separating P and L in this way is to think of P as independent of $O_3$ and L as being linearly proportional. Unfortunately, while the P reactions (2) have no obvious $O_3$ terms, both these reactions and the OH and $HO_2$ abundances in reactions (3) depend indirectly on $O_3$; and thus with a true linearization of P–L, the lifetime of $O_3$ is much shorter than inferred from L (Prather and Holmes, 2013). A similar

chemical feedback with opposite sign occurs for $CH_4$ whereby the lifetime of a $CH_4$ addition is longer than inferred from the linear relationship of reaction (1) (Prather, 1996). We retain these definitions of P-O3, L-O3, and L-CH4 because they still represent the reactivity in remote regions and the reaction rates, rather than a linearization, are straightforward CTM/CCM diagnostics.


We define the reactivity of an air parcel (reactions 1-3) in terms of 24-hour average rates and hence the units of ppb per day. Reactivity defined here requires sunlight; nighttime sources of OH from alkenes and isoprene via ozonolysis or nitrate radicals (Paulson and Orlando, 1996) are important primarily in continental air over emission sources. This calculation integrates over the

diurnal cycle of photolysis rates driven by changing solar zenith angle, clouds, $O_3$ and aerosol profiles, all of which are simulated in CTM/CCMs.

What key constituents are needed for modeling reactivity? Models simulate many tens to hundreds of chemical species. While many are important for calculating the instantaneous

reaction rates, e.g. $O(^1D)$, they are not the key species. Key is defined here as a constituent whose initial value significantly affects the 24-hour reactivity, whereas other species can be initialized to any reasonable value and not affect it. For example, OH and $HO_2$ are radical HOx species whose abundances directly determine the rates of reactions (1-3). Nevertheless, these are not key species as their abundances can be initialized to zero and are rapidly reset in seconds to a





temporary steady state with first sunlight or changing clouds through reactions 3cd among others (Rohrer and Berresheim, 2006). This argument applies to similar radical species such as $CH_3OO$, but not to HOx sources like $CH_3OOH$ and $HOOH$ whose initial values will control the abundance of OH and the reactivities over the day.

A similar situation applies to NO and $NO_2$ (collectively designated NOx), whereby total NOx changes over the day as it is exchanged with higher oxides of nitrogen but the fraction of NOx in the form of NO is determined rapidly in sunlight by reactions 2abc and 4.

$$NO + O_3 \rightarrow NO_2 + O_2 \qquad\qquad\qquad (4)$$

In the dark, NOx is almost entirely $NO_2$, and it is critical to initialize NOx, but not NO and $NO_2$

separately.

Based on sensitivity tests with the UCI CTM, our list of 18 key species includes:

$O_3$, NOx, $HNO_3$, $HNO_4$, PAN ($C_2H_3NO_5$ = peroxyacetylnitrate), $RNO_3$, ($CH_3NO_3$ and all alkylnitrates), HOOH, ROOH ($CH_3OOH$ and smaller contribution from $C_2H_5OOH$) ,

HCHO, $CH_3CHO$ (acetaldehyde), $C_3H_6O$ (acetone), CO, $CH_4$, $C_2H_6$, alkanes (all $C_3H_8$ and higher), alkenes (all $C_2H_4$ and higher), aromatics (benzene + toluene + xylene), $C_5H_8$ (isoprene + terpenes).             (5)

We also add p (hPa), T (K), q ($g-H_2O$/kg-air), and latitude and longitude to make up the 23 key variables in each air parcel. Some collectives like alkanes may be treated as multiple, separate

species in some models, or may be lumped according to their reaction rates. List (5) tends to be inclusive because for much of the troposphere, a smaller list can apply. For some species (e.g., $C_5H_8$), their role is key only if they are present in large enough abundances, but even when sampling across the Pacific Ocean basin one may find plumes with recent biospheric sources.

This simplification of the chemical system fails in regions of intense emissions of short-lived species or in highly polluted environments such as urban, industrial, or open fires. After pollution plumes have been separated from sources and aged a few days, then our key variables should define the reactivity. Such conditions apply to most of the troposphere, particularly the air over the vast Pacific and Atlantic Ocean basins. With aged pollution plumes, we expect that

some key species (e.g., alkenes, isoprene, aromatics and higher alkanes) will drop off the list



because their abundances in much of the remote mid-ocean regions will have fallen below the relevance threshold.

### 3. Modeling the reactivity of air parcels


Why use the global models instead of single-box models to calculate reactivity statistics? There are several reasons. For one, these CTM/CCMs simulate the full meteorology including cloud cover and its variation over large regions, which is a critical component of reactivity. Second, they usually include self-evaluated ozone and aerosol profiles also needed for the photolysis

rates. Third, these models automatically simulate the diurnal cycle in radiation at all seasons, latitudes and longitudes. And fourth, most importantly, these models have built-in chemistry modules that already calculate reactivities, and they are the ones we rely on for climate and air quality assessments. The goal here is to test their simulated chemical heterogeneity. While a box model could be designed (using 3D meteorology) to address the first three needs (e.g.,

Nicely et al., 2016), it cannot address the last. More simply, all the necessary Earth system components are already built in to the CTM/CCMs, and our approach of testing the modeled climatologies includes that of testing the Earth system components (e.g., emissions, transport, chemistry, scavenging, air-sea exchange, and land-surface interactions).

In a standard CTM or CCM simulation (defined here as a C-run) we calculate the reactivity at a given grid cell, but not that of a parcel. Air parcels move, change location, and mix with neighboring parcels: i.e., there is no way to track quantitatively what might be considered the original parcel. Effectively, we keep integrating the rates in that grid cell as different parcels travel through it and are mixed within it. Let us take a large enough domain of grid cells (e.g.,

tropical Pacific, 150E to 210E, from surface to 200 hPa) and calculate the statistical distribution of reactivities of all those grid cells. We take these statistics to be equivalent to those we would get from integrating the reactivity over isolated air parcels with the same initialization. Of course the latter is only a thought experiment since the parcels do not remain isolated. In C-runs new air parcels are entering the domain and others are exiting. In a single cell we can start with

a polluted lamina and end with clean air convected from the marine boundary layer, but much of the polluted lamina remains in the larger domain. As long as the domain retains a statistical mix



of the key chemical species similar to the initialization, then the reactivity statistics of the C-run should represent the hypothetical reactivity of those initialized parcels.

How can we design a calculation using the CTM/CCMs that allows us to initialize a subset of grid cells with observed air parcels and then calculate a reactivity for those parcels? The goal here is to be able to use the NASA ATom aircraft mission (2015-2020), which was designed to measure those 23 key variables in air parcels profiling from near-surface to 12 km altitude, flying ascents and descents down the middle of the Pacific and Atlantic Ocean basins. Thus ATom

data will not fill the global 3D model grid, and thus many cells will be initialized with the model's original chemistry values. The critical design requirement is that we let the model integrate for 24 hours as it normally does in a way that the chemistry in each grid cell depends minimally on any of the grid cells around it.

We thus propose an A-run mode (named after the ATom mission) for the CTM/CCMs in which individual parcel reactivities can be calculated, albeit with some simplifying approximations. Consistent with our definition of reactivity, we consider only ATom parcels that are tropospheric. The A-runs disable processes that connect and mix air parcels. First we drop all calls to the tracer transport sections (advection, convection, diffusion, boundary layer mixing).

Second, we must cut all emissions, including lightning and aircraft NOx, because without transport, the emissions would build up unrealistically in the source cells. Third, all tracer scavenging modules must be turned off because in many models the scavenging depends on the vertical distribution of the species.

In this A-mode, the remaining connection of the reactivity calculation with neighboring grid cells is through the photolysis rates, which require profiles of clouds, aerosol layers, and ozone. It is impossible to prescribe all these data over the diurnal cycle for each parcel from observations, and thus we must rely on the CTM/CCM to generate a suitably realistic, diurnal, regional, seasonal climatology for these and hence the photolysis rates. To better average the reactivity

over synoptic variations in clouds, we expect to repeat the same initialization of the A-runs for a range of days over a month containing the observations.



Each ATom parcel (2-8 km along the flight path) will be assigned a unique model grid cell to best match the observation: latitude and pressure grids containing the measurement, and

longitude chosen as close as possible but maintaining a unique cell for each parcel. ATom parcels in adjacent grid cells may represent air masses separated by a few km instead of the grid-cell size of order 100 km. A high density of ATom parcels in a region will be placed in the correct latitude and pressure cells but may be strung out in longitude cells. The parcel will use the mole fraction of key species, water vapor (q) and temperature (T) as measured, but will adopt

the mean pressure of the grid cell. The model may need to maintain separate storage for the hourly T and q used in the CCM dynamics because it is important to maintain the clouds as they would be done in the C-run, and thus the main-code values of T and q cannot be overwritten with ATom values. The A-run treatment of stratospheric $O_3$ (i.e., fixed) is unlikely to be identical to the C-run, but it does not appear to drive major changes in the average photolysis rates over a

region (see below).

In defining the A-runs thus, we have created some biases in the reactivities relative to the C-runs. Examination of the NOx and HOx budgets of parallel A- and C-runs shows two obvious differences. The A-runs lack emissions and over the remote ocean basins, where the most

important emission is NOx (lightning, shipping, aviation). Thus A-runs show a 24-hour decline in NOx abundances compared with the C-runs, resulting in generally lower P-O3. The A-runs also lack scavenging, and thus accumulate more $HNO_3$ and HOx precursors like HOOH, affecting L-CH4. No other simple objective approach has been found, and we must accept and document these biases in the A-runs.


An examination of how the A- and C-runs differ is shown in Figure 2 using the UCI-CTM's 1D probability distributions of six key species (NOx, $HNO_3$, HNO4, PAN, HCHO, HOOH) for the central tropical Pacific. The initial distribution for both runs (12h local solar time at 180E, black solid) can be compared with that 24 hours later (36h) for the C- (black dashed) and A-runs (cyan

squares, only for 4 species). The number of moles at the beginning and end of the 24-hours in the C-run (see legend) is a measure of the daily changes in the air parcels entering and leaving the domain. It varies from 0 to 4%, well within the expected representativeness of a given day. With the A-run, however, we see large systematic shifts due to the lack of emissions (NOx) and



scavenging (HNO$_3$, HOOH). For HNO$_3$ the content increases overall by 9%, with the high-end
(>100 ppt) distribution not changing but the low-end (<20 ppt) air gains HNO$_3$, increasing the
middle section (20-100 ppt). This is logical because the low-HNO$_3$ regions have the most
scavenging. This change in distribution over the 24-hour integration of the A-runs is unlikely to
change the reactivities as the release of NOx from HNO3 will be more important in the high-
HNO$_3$ regions. For NOx the content decreases overall by 18%, with most air parcels (4-100 ppt)
becoming less frequent and an increase in frequency only for parcels with very low NOx, <4 ppt.
The 1D distribution of HCHO shifts lightly, but with little overall change in content. The lack of
scavenging is even more important for HOOH with an overall increase of 41% and a dramatic
shift in the distribution: decreases in 0.3-1.0 ppb appearing as very large increases from 1.0 to
2.5 ppb. The implications for using the A-run bias in computing the reactivities is examined with
all 6 models below.

An important assumption in using key species to initialize the reactivity simulations is that the
diurnal cycle is not critical, and that ATom measurements can be used without trying to make
corrections for the time of measurement. In running these global models, it is not practical to
initialize parcels at other than a standard day (i.e., beginning at 00h UT). For some species like
HCHO, the daytime loss frequency in the tropics is about 1/2 hr$^{-1}$ (see for example loss
photolysis rates for various oxygenated hydrocarbons in Prather (2015)), and thus one might
expect it to vary greatly over the sunlight day or with cloud variations. The diurnal change in 1D
distributions of the 6 key species is also shown in Figure 2 for the C-runs at 18h (local solar time,
red dashed), 24h (dark blue dashed) and 30h (green dashed). The C-runs are in approximate
steady-state over the tropical Pacific domain as seen by comparing 12h with 36h, and thus these
sunset-midnight-sunrise times show the daily variations. The diurnal cycle does produce visible
shifts in the 1D distributions, particularly at the end of night (30h). The shifts in HCHO are
small considering its high loss frequency, primarily because both sources and sinks respond
similarly to photolysis rates. The seemingly longer-lived HOOH shows larger shifts because
production occurs in sunlight but scavenging occurs day and night. PAN and HNO$_4$ show small
diurnal cycles at the high-abundance end of their distributions where they can be important NOx
sources, and initialization errors caused by the diurnal cycle at the low-abundances will have
smaller impacts on reactivity.




A test of A- versus C-runs for all 6 CTM/CCMs is shown Figure 3. All models were spun up for a year and stopped at 00H UT on 16 August, with the chemical abundances at this time being used to initialize each model's own C- and A-runs. In this case all species in the model were initialized and not just the 18 key species. Each model ran their own chemistry and meteorology

intended to simulate a specific historical year or a typical climate year. All were intended to be typical of the last decade. The models were then run 24 hours and the rates and reactivities diagnosed for both C-runs and A-runs. All models have different resolutions, ranging from 0.5 to 2 degrees. All model statistics (key variables, reactivities, plus 24-hour average photolysis rates) were stored globally. This analysis examines a north-south transect flight over the Pacific

Ocean basin as in the NASA ATom flights, but greatly expands the region to include more grid cells: latitude, in 6 domains 60S-40S-20S-0-20N-40N-60N (each region is color keyed in Fig. 3); longitude, in a single broad domain 150E-210E. Vertical profiles (200 hPa–1000 hPa) on the models native grid are shown for the 6 domains as different colors. The standard C-runs with all transport and emissions included are solid lines, while the ATom-like A-runs are dashed.


For L-CH4, the only general agreement is the lesser importance of parcels at altitudes above 500 hPa. For this August test, most models find that the 20N-40N dominates (note that plots are ppb/day and not area weighted), and the 60S-40S and 40S-20S domains are least important (similar to OH structures in Spivakovsky et al. (2000); Lawrence et al. (2001)). Most models

show increasing L-CH4 in the first few km above the ocean because of low-level clouds shifting photolysis to the middle troposphere   The results for L-O3 show similar patterns of agreement and disagreement among models but emphasize the dominant role of the middle troposphere (500–800 hPa) for $O_3$ loss. P-O3 has distinct patterns, demonstrating the importance of larger NOx values in the upper (200–500 hPa) and lower troposphere (800–1000 hPa), presumably

from lightning NOx. Only GMI-CTM lacks lower troposphere sources of $O_3$ about 180E. Overall the models show modest, similar amplitudes (but not always sign) in the bias of A-runs relative to C-runs. Thus we can use the model A-runs to tag each parcel in the ATom measured climatology by its reactivity in the absence of emissions and transport. Clearly these models have largely different chemical climatologies for the middle of the Pacific, and with the ATom



climatology to initialize all six models, we will be able to test if these differences reflect the
initial key species and/or the photochemical components.

Photolysis rates (J-values) are the driving force for reactivity, and we include also a comparison
of the 24-hour average J's (reactions 2d and 3c) in Figure 4. The model spread in J-$NO_2$ is 20%
and likely due to differences in cloud cover as well as the photolysis module in the model. The
wide, factor-of-two range in J-$O_3(^1D)$ cannot be simply explained through differences in clouds
or ozone, for example, a 20% reduction in column $O_3$ gives only a 33% increase. Such
differences will drive a large part of the model differences seen in Figure 3. For example, the
large J-$O_3$ for GISS, and hence large production of OH, can explain in part why GISS has very
large L-O3 and P-O3, but not why the L-CH4 (also dependent on OH) matches the other models.
Surprisingly GEOS-Chem has even larger J-$O_3$ but its reactivities are within the range of the
other 4 models. A comparison between the A- and C-runs (not shown) confirms that these two
runs have almost identical J's as expected since these changes in ozone and aerosols over 24
hours between these two simulations will have small impact on regional average J's.


While the A-run is clearly asking the modeling groups to make some rather uncomfortable code
modifications, these tend to be at the very high level of disabling entire components. We
recommend this approach as it will allow us to more directly compare modeled reactivities,
including when all models are initialized with the same chemical composition.






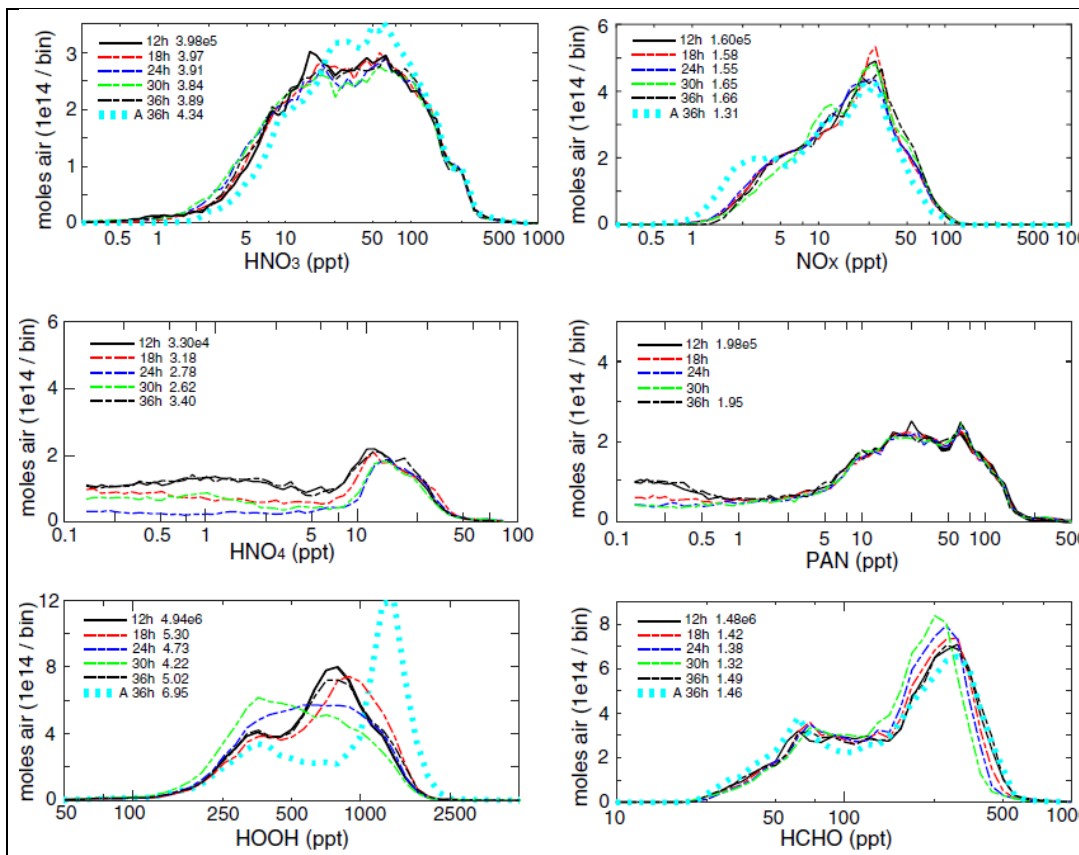

Figure 2. 1D probability distributions for HNO$_3$, NOx, HNO$_4$, PAN, HOOH, and HCHO from the UCI-CTM. The domain sampled is the tropical Pacific: 20S-20N, 150E-210E, 0-12km, on 16 August. The units are moles of air per log-scale bin (20 bins per factor of 10). The area under the curve in the log plot is the air mass of the domain, except for HNO$_4$ and PAN for which there are numerous observations below the cutoff at 0.1 ppt. Five different times are shown for the C-run: local noon (12h), sunset (18h), midnight (24h), sunrise (30h) and the following noon (36h). Also shown is the A-run at noon (12h, same as C-run) and the following noon (A 36h). The numbers of moles of the species in the domain are given in the legend.







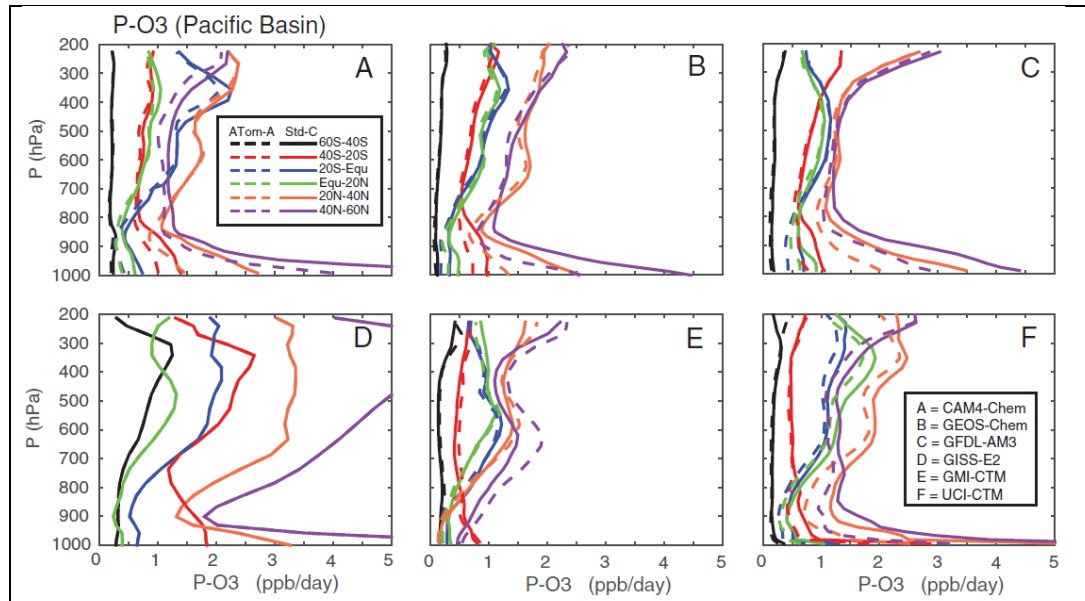

Figure 3. Profiles of reactivity (ppb/day) for loss of CH4 (L-CH4, top panel), loss of O3 (L-O3, middle panel), and production of O3 (P-O3, bottom panel) from 6 global models (Table 1). Cells from each model grid are averaged over 20-deg latitude domains (different colors, see legend), longitudes from 150E to 210E, and for the single day of 16 August. Years vary by model, see text. Solid lines are standard model simulations (C-runs) with the values representing air that passed through the cell over 24 hours. Dashed lines are the no-transport, no-emissions A-runs that keep the initialized chemical values in the same cell over 24 hours.


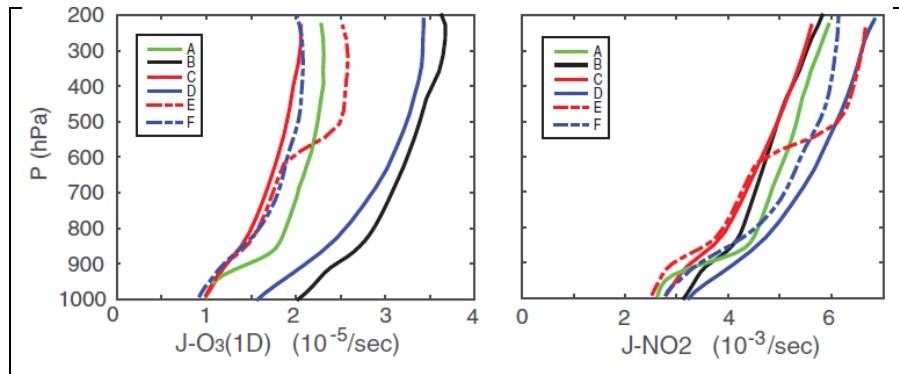

Figure 4. Modeled 24-hour average J-values for $O_3 + h\nu => O(^1D) + O_2$ (left, $s^{-1}$) and $NO_2 + h\nu => NO + O$ (right, $s^{-1}$) for the tropical Pacific (20S-20N, 150E-210E). See Fig. 3 and Table 1 for model codes.



### 4. Probability distributions of species and reactivities

We characterize the heterogeneity in tropospheric chemistry through the joint-probability

distributions of the frequency of occurrence of chemical species in air parcels for the six models

here. These diagnostics are readily suited to high-frequency in situ observations from an

extensive aircraft mission such as ATom, for example see (Koppe et al., 2009). This paper then

takes a novel approach by focusing on the chemical budgets for tropospheric ozone and methane.

In addition to weighting a parcel according to its occurrence or parcel mass, we include a factor

that accounts for the model-calculated reactivities of that parcel. For example, the basic weight

of a parcel (moles-air) can be scaled by P-O3 (ppb/day) and the final weight is the moles-$O_3$/day.

In this case the sum of weighted parcels in a region gives the moles of $O_3$ produced per day in

that region. These reactivities can be calculated with A-runs for both models and

measurements. Thus, the modeled and measured probability distributions reflect the parcels

most important in determining the chemical budgets in these models, and hence the evolution of

the atmosphere.

Given the number of key species, the joint probability distributions are multi-dimensional, but

for the most part we view them 1D or 2D graphs. There is a history of comparing models and

measurements using such graphs (Hoor et al., 2002; Hsu et al., 2004; Engel et al., 2006; Pan et

al., 2007; Strahan et al., 2007; Parrington et al., 2013; Gaudel et al., 2015). Often the goal is

simply to define a linear correlation, but in many cases a line-fit simply does not describe the

heterogeneity (Koppe et al., 2009).


A much more difficult problem is that of representativeness: i.e., How much of the Pacific basin

must one sample to get joint probability distributions similar to that of the whole basin? Can

aircraft-measured heterogeneity be compared with models that do not follow the exact flight

route for the exact period of measurements (e.g., Hsu et al., 2004)? This latter question is critical

if we are to use the ATom measurements to test such a wide variety of CTM/CCMs. Here, we

consider an idealized test case for representativeness where we sample a model as objectively as

possible and then compare with different sampling 'paths.'





One test of representativeness looks at the reactivities sampled along a single longitude and then
integrated over latitude-pressure domains.  For example, Figure 1a clearly shows that the
instantaneous column integrated L-CH4 varies greatly along longitude transects in the mid-
Pacific.  The point-to-point variance in 3D will be very large, but if we average over regional
domains, can we achieve a representative mean value for reactivity?  Based on the profiles of
reactivity (Fig. 3) we take three pressure domains (surface – 850 hPa – 500 hPa – 200 hPa, but
with stratospheric values screened out by model-designated discriminators) and three latitude
domains (60S – 20S – 20N – 60N).  The means (ppb/day) and standard deviations (ppb/day) of
single-longitude sampling across the mid-Pacific (155E – 233E) on 16 August are shown for the
UCI-CTM in Table 2 along with the standard deviation (in %) over the 31 days of August of the
daily full-domain average.  The standard deviations are a measure of the representativeness of
the sampling, by longitude or by day.  For L-CH4, the dominant mean loss, >1 ppb/day, is in the
surface – 500 hPa in the tropics and summer (northern) mid-latitudes as seen in Fig. 3.  For these
regions the standard deviation across the longitudinal samples is of order 6-11%; whereas
outside of these, it is as large as 20%, but the absolute values are small.   A similar pattern holds
for L-O3 with standard deviations in dominant regions of 6-14%.  Thus any single, fully sampled
longitudinal transect through this domain has a 68%-likelihood of being within 6-14% of the
mid-Pacific average.  The variance of P-O3 is slightly larger, 8-17%, in part because P-O3
depends on the less-frequent high-NOx regions.  Assembling a representative sampling of P-O3
at the same % level as L-O3 will be slightly more difficult.  Such single-transect
representativeness is about as good as we can expect.  Thus, model-model differences comparing
individual transects from each model would not be significantly unless they exceed these
percentages.  Averaging over the basin and/or several days should resolve model differences at
finer scales.  The day-to-day standard deviation for the mid-Pacific averages in Table 2 is shown
in percent; it is smaller than across individual longitudinal transects for a given day; and in key
regions (surface to 500 hPa, 20S to 60N) it ranges from 1-4% for L-CH4 to 2-8% for P-O3.  A
remaining question (not resolved with the data sets assembled here) is the year-to-year variance
of basin-wide reactivities perhaps associated with the El Nino – Southern Oscillation.

The six models' 1D probability distributions for O₃, CO, NOx, and HCHO over the tropical mid-
Pacific basin are shown in Figure 5.  Modeled data is sampled on the native grid of each model





and not interpolated.  This approach readily allows us to compare different models.  Both 1D and

2D distributions presented here are sorted into 20 log-spaced bins per each factor of 10 (decade)

in abundance (ppb or ppt).  The dashes in the upper/lower rows of Fig. 5 indicate widths of these

bins on each plot.  For example, NOx distributions cover more than 3 decades (very small

dashes), while the CO covers less than a decade (wide dashes).  In the first row labeled "AIR",

each grid cell is weighted by its size in moles, and thus the plot shows Petamoles per logarithmic

bin.  In each subsequent row, the cells are weighted by the reactivity (L-CH4, L-O3, P-O3) in

moles/day, plotting thus Megamoles per day per bin.

The AIR plots show clear model differences.  Models A and B have much greater frequency of

$O_3$ occurrence from 50 – 150 ppb, and half the models (B, D, E) show a reasonable frequency of

$O_3$ at 10 ppb and less, as might be expected in the tropical Pacific boundary layer (Kley et al.,

1996; Singh et al., 1996; Nicely et al., 2016).  For CO, model A shows unusually low

abundances.  For NOx, models C and F lack the NOx below 2.5 ppt that others have.  The

models are quite similar for HCHO, except for D, which has an unusually symmetric distribution

and much lower abundances.  When reactivity weighted, new features are found.  Note that the

area under the AIR-weighted curve is the same for all models, but the area in reactivity-weighted

1D plots is each model's total reactivity (moles/day).  Model D has lower values overall for L-

CH4 compared with the other models, but it is similar or even slightly higher for L-O3 and P-O3.

The high-O3 abundances in A remain equally important when weighted by any reactivity, but

those in B become less important for L-CH4 and L-O3, but even more important for P-O3.  This

unusual feature adds a new dimension to diagnosing and understanding model differences.  The

reactivity weighting of the CO distribution does not show anything unusual.  The NOx 1D plots

show that L-CH4 is more heavily weighed to low NOx values than is L-O3, but P-O3 is

weighted strongly to the higher NOx abundances (>10 ppt) as expected.  The HCHO reactivity

weights in the opposite direction with high abundances (>200 ppt) favoring L-CH4 and L-O3 but

lower ones favoring P-O3, probably because the lower ones are from the upper troposphere

where colder temperatures suppress both L-CH4 and L-O3 (Fig. 3).  The reactivity weighting

adds a new dimension to the diagnostics, and after the ATom data set becomes available it would

be productive to make a more detailed comparison that identifies the location and other key

species controlling these shifts in reactivity.



These new diagnostics do not instantly identify the cause of model differences, but they do add a new dimension. For example, if we seek to understand why model D is different, we can look at global budgets: both models A & D have P-O3 and L-O3 tropospheric means between 2.5 and 3.5 ppb/day; whereas the other 4 models have values between 1.0 and 2.0. The global L-CH4 -- 0.50 to 0.65 ppb/day -- is similar for all models, with D in the middle. So globally models B & D are similar, but in the mid-Pacific, they are distinct with model D having much lower L-CH4 values in the tropics and especially the lower tropics (Fig. 3, see also Fig. S1 of Naik et al., 2013b). CH$_4$ loss is a major source of HCHO in the unpolluted atmosphere and this may partly explain D's lower values of tropical HCHO compared with other models. Some of the reduced tropical reactivity in D may be caused by more low clouds in the tropics, and this is apparent in the more rapid fall off in J-O$_3$($^1$D) compared with other models (Fig. 4); yet models B & D (not A & D as found in L-O3 and P-O3) have much higher values of J-O$_3$($^1$D). With the ATom A-run approach we will be able to remove differences caused by the widely ranging chemical climatologies of species (e.g., seen in Fig. 5, 6, 8) and more directly trace the range of results to the models' basic photolysis and kinetics.

The 2D distributions simply weighted by AIR show remarkable structures that differ significantly across the models, as shown in Figure 6. All 2D plots use the same 20-per-decade log scale as in the 1D analysis, and they are normalized such that if all parcels are distributed uniformly within a 20x20 square (e.g., 0.1-1.0 ppb HOOH, 10-100 ppt NOx) the arbitrary density value would be 1 (a yellow-green color in Figures 6-7). Thus the reactivity-weighted 2D plots are renormalized and do not reflect the individual model's total reactivity. In Figure 6a the AIR-weighted NOx-HOOH plots show a boomerang structure with greatly varying degrees of concentration about some points in the center (reddish regions). For example, models A and D show a very diffuse distribution with a much wider spread in HOOH values at lower NOx. Even for the four models with a central (NOx, HOOH)-line defining a peak frequency of occurrence, this line occurs at different locations. The O$_3$-H$_2$O density plots (Fig. 6b) show examples of highly standard and well-measured species with extreme distributions: O$_3$ fall within 1 decade throughout most of the troposphere, but H$_2$O easily spans 3. Several show the bimodality of many parcels with low O$_3$ with high H$_2$O (marine boundary layer and above) and a





second peak at higher $O_3$ and dry. For example, C and E look very much alike, but B has these two peaks more separated, and E has a much broader spread in upper tropospheric $O_3$ abundances.


These plots are only tropical but include altitudes that can be sampled by the ATom flights. When comparisons with ATom data are made, it will be useful to identify discrepancies in the 2D plots with altitude or other features.

The 2D plots can change the emphasis of certain regions when weighted by reactivity. For example, we take the GMI-CTM modeled NOx-HOOH density (Fig 6a panel E) and show the reactivity weightings in Figure 7. With AIR weighting, the quasi-boomerang has a strong central line with a negative slope. With P-O3, a much broader range is seen and the peak occurrence shifts to lower HOOH values and somewhat even to lower NOx values. With L-CH4, the line

disappears and a galaxy-like pattern widens the range of parcels, picking up lower NOx values in two spiral arms. The L-O3 weighting is similar to L-CH4, and differences are discernable only in small features. Clearly, species other than NOx and HOOH determine the reactivity of parcels, and thus other 2D plots will add new information. We anticipate that ATom measurements will be plotted not only with AIR weightings but also with reactivities calculated

for that air parcel with these models (Auvray et al., 2007).

The 2D plots shown here intentionally included all air parcels over the mid-Pacific to ensure that a robust distribution was obtained (see Table 2). If we have only a single longitude slice as in ATom, then will these be so clearly defined? We examine this representativeness test by sub-

sampling two models (C: GFDL-AM3 and F: UCI-CTM) at longitudes of 150E, 165E, 180E, 195E, and 210E in Figure 8 to compare with the average over the mid-Pacific domain. The densities are renormalized and show similar peaks and patterns, but of course there is more pixel-level noise and some differences. The transect at 150E is clearly less representative of the mid-Pacific, which is understandable since that longitude includes Papua New Guinea and eastern

tropical Australia. Most importantly, the differences for 165E-210E are less than those across the 6 models (Fig. 7a). We need to develop an objective measure for comparing 2D plots





between models and ATom measurements, and for judging if their differences are within the range of representativeness.

Table 2. Representativeness of reactivities (L-CH4, L-O3, P-O3 all in ppb/day) averaged over 3 latitude and 3 pressure domains over the central Pacific (155E-233E). The first standard deviation (ppb/day) is over the different longitudinal transects on mid-August; and the second (%) is for the average across longitudes sampled over 31 days of August.

| L-CH4 (ppb/day) | 60S-20S | 20S-20N | 20N-60N |
|---|---|---|---|
| 500-200 hPa | 0.08 ± 0.02 ± 8% | 0.36 ± 0.06 ± 7% | 0.45 ± 0.08 ± 3% |
| 850–500 hPa | 0.28 ± 0.04 ± 6% | 1.08 ± 0.07 ± 4% | 1.26 ± 0.12 ± 2% |
| surf–850 hPa | 0.35 ± 0.03 ± 4% | 1.21 ± 0.13 ± 2% | 1.44 ± 0.11 ± 1% |
| L-O3 (ppb/day) | | | |
| 500-200 hPa | 0.24 ± 0.03 ± 7% | 0.74 ± 0.15 ± 8% | 1.35 ± 0.27 ± 4% |
| 850–500 hPa | 0.69 ± 0.08 ± 6% | 2.28 ± 0.13 ± 6% | 3.01 ± 0.43 ± 3% |
| surf–850 hPa | 0.86 ± 0.06 ± 4% | 2.46 ± 0.32 ± 3% | 2.60 ± 0.22 ± 3% |
| P-O3 (ppb/day) | | | |
| 500-200 hPa | 0.35 ± 0.04 ±10% | 1.37 ± 0.18 ± 7% | 1.78 ± 0.30 ± 4% |
| 850–500 hPa | 0.36 ± 0.06 ± 9% | 0.92 ± 0.08 ± 8% | 1.46 ± 0.13 ± 2% |
| surf–850 hPa | 0.32 ± 0.20 ± 9% | 0.43 ± 0.09 ± 2% | 2.34 ± 0.33 ± 3% |

Results are from the UCI CTM C-runs for 16 August and 1-31 August. The 155E-233E domain includes 69 longitudinal transects. All grid cells in the domain are sampled equally, but in stratospheric parcels the reactivity is not included. The period 1-31 August shows trends in some domains as the sun moves southward, and this was removed with a line fit to calculate the standard deviation over the month. Results for the A-runs (not shown) differ in mean and standard deviation by a few percent.


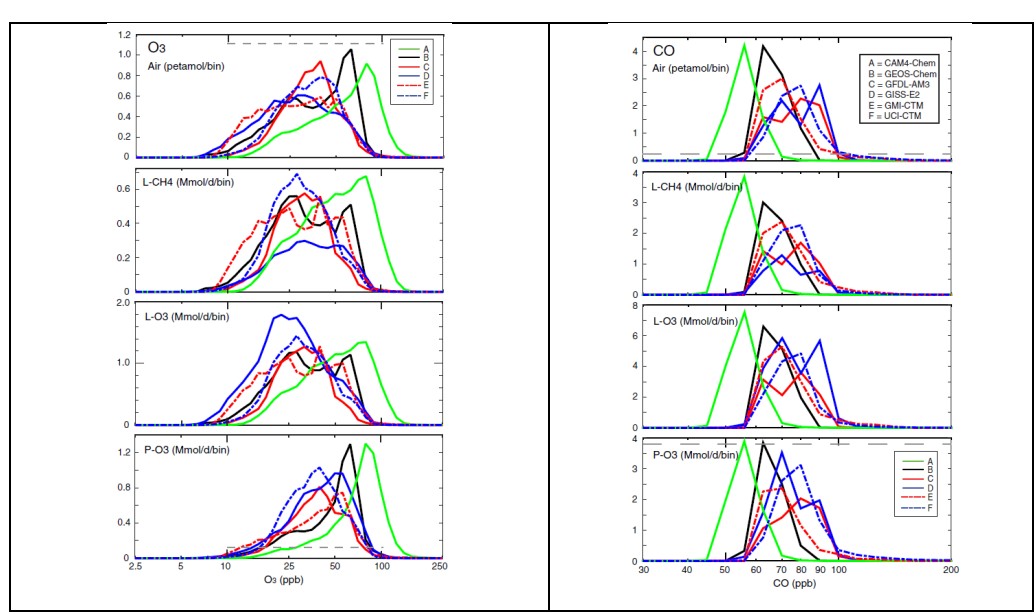



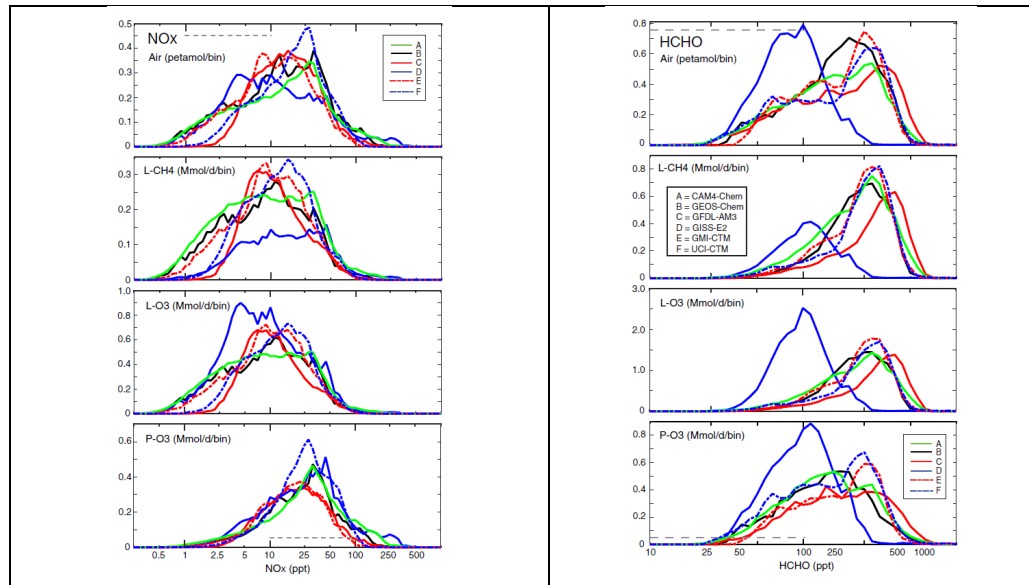

Figure 5. Six modeled 1D probability distributions for O₃, CO, NOx, and HCHO, where the air parcels have been weighted by air mass (row 1), L-CH4 (row 2), L-O3 (row 3), and P-O3 (row 4). The domain being sampled in the models is the tropical Pacific: 20S-20N, 150E-210E, 0.5-12km. Units for the air-weighting are petamoles per bin where the bins are set at 20 per decade (sizes marked by dashed lines in upper or lower panels) and Mmoles per bin per day for the reactivity weighted plots (rows 2-4).





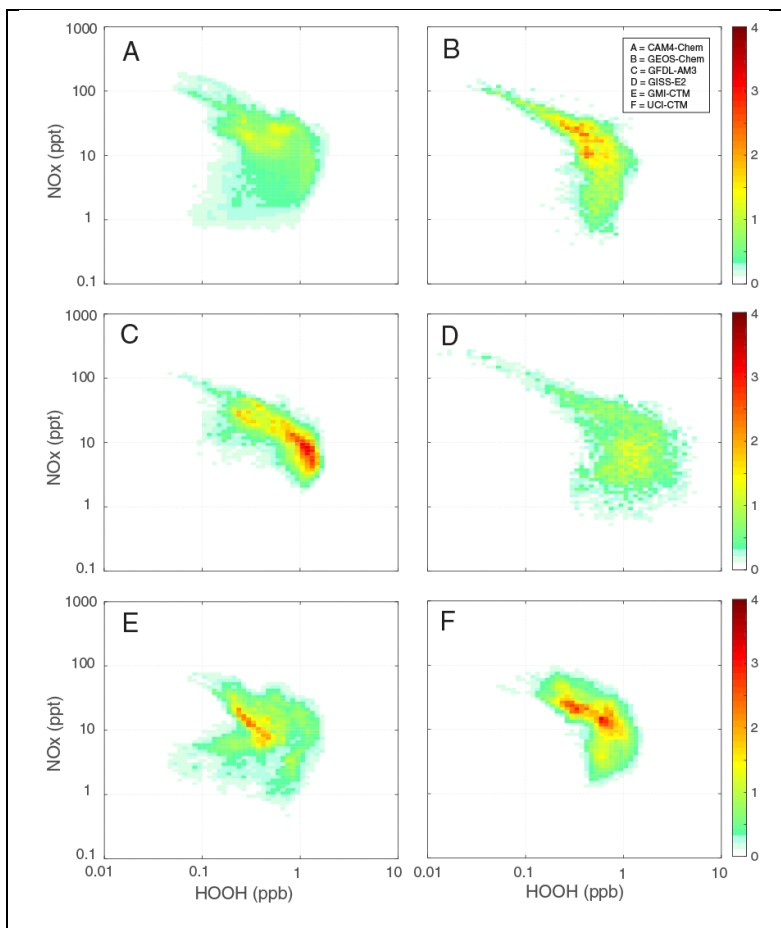

Figure 6a. Six modeled 2D probability distributions for NOx vs. HOOH as weighted by air mass. These are the initial chemical abundances for each model and hence the same for A- and C-runs. All grid cells were binned at 20 per decade in species abundance (mole fraction, ppt for NOx, ppb for HOOH). The density value for each plot is scaled so that a uniform distribution over exactly one decade in both species would give the yellow-green color of 1.0. The domain being sampled in the models is the tropical Pacific: 20S-20N, 150E-210E, 0.5-12km. Model A = CAM4-Chem; B = GEOS-Chem; C = GFDL-AM3; D = GISS-E2; E = GMI-CTM; F = UCI-CTM.





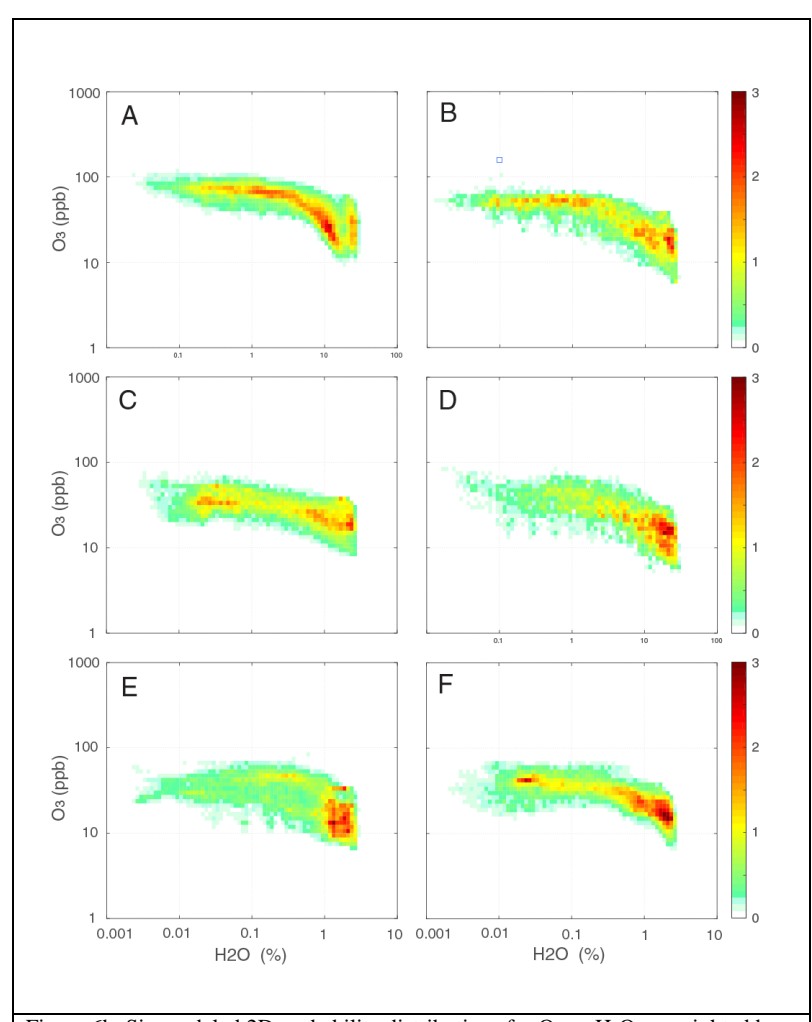

Figure 6b. Six modeled 2D probability distributions for $O_3$ vs $H_2O$ as weighted by air mass. This color-bar scale differs slightly from other 2D plots. See Fig 6a.






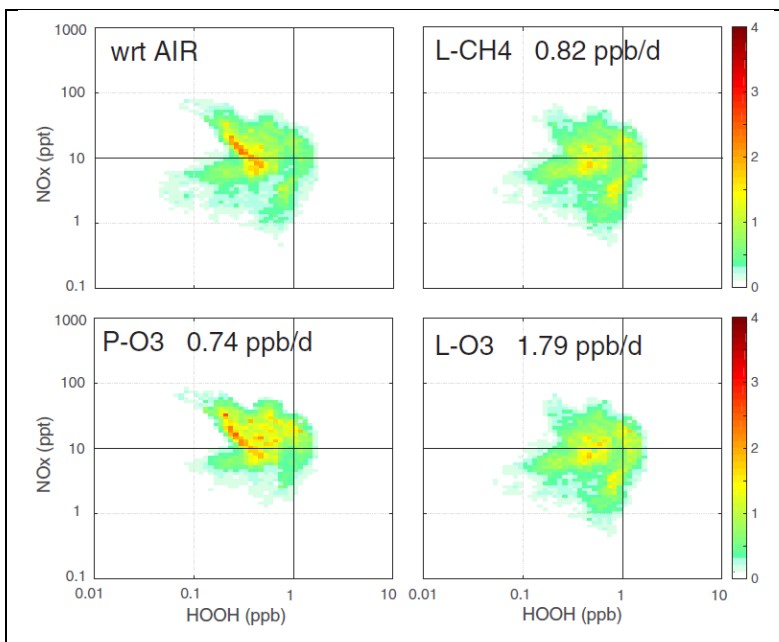

Figure 7. Model E (GMI CTM) 2D probability distributions from A-run for NOx vs. HOOH as weighted by air mass, L-CH4, L-O3 and P-O3. The domain being sampled is 20S-20N, 150E-210E, 0.5-12km, see Fig 6a.



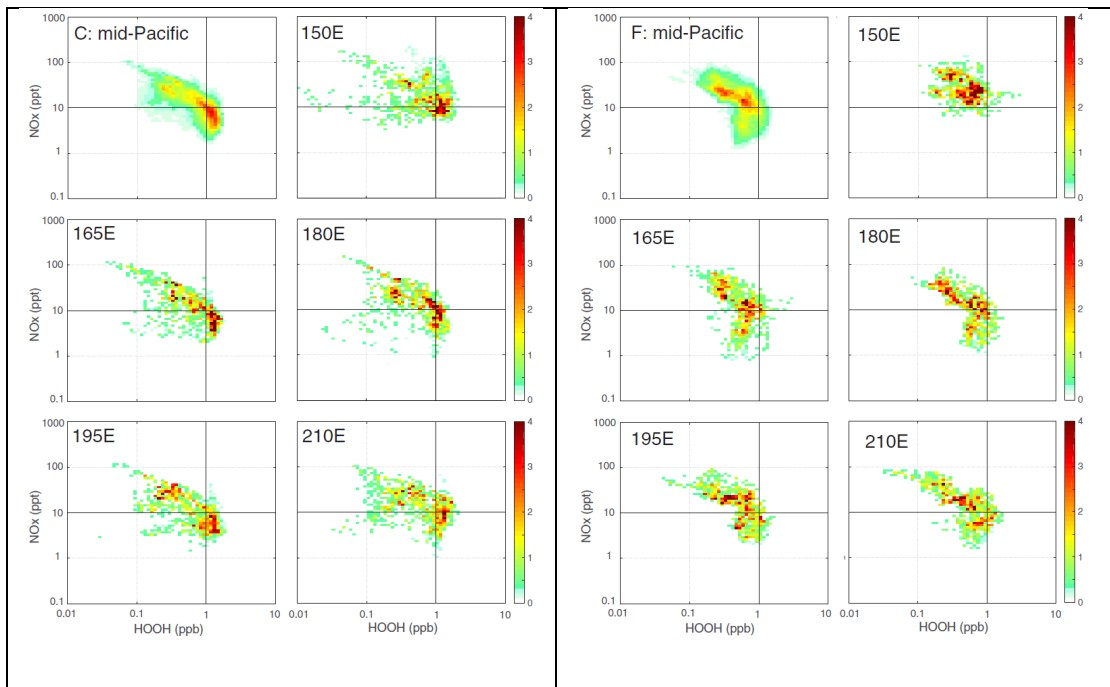

Figure 8. AIR-weighted 2D probability distributions for NOx vs. HOOH averaged over mid-Pacific (150E-210E) and at different single longitude transects from 150E to 210E, shown for models C (GFDL-AM3) & F (UCI-CTM). The region sampled is 20S-20N, 0.5-12km.





*5. Discussion and preparation for the ATom data set*

This paper is based on the underlying premise of the NASA ATom missions – that high-
frequency measurements of the key species controlling the daily-average reactivity of individual
air parcels can be made on regular profiling flights through the middle of the Pacific and Atlantic
Ocean basins, and that these air parcels will provide a unique, objectively sampled chemical
climatology identifying those air parcels that are most important in controlling methane and
tropospheric ozone. This data set will provide the most rigorous testing and diagnosis of the
global chemistry models, in particularly the chemistry-climate models, which require a
climatology, not a campaign data set.


Here we have outlined the model development that will enable the global chemistry models to
readily use the ATom measurements and calculate the chemical reactivity in the parcels (i.e., the
A-runs). Thus once the ATom measurements have finished quality controls and a coherent data
set produced, the six models here can readily calculate the production and loss of ozone, the loss
of methane, and even the possible gas-phase sources of aerosols in the remote troposphere as a
valued-added ATom data product to accompanying the measurements.

The 1D and 2D probability distributions, shown here as a sample, are sufficiently diverse across
the models that ATom measurements will clearly be able to differentiate among them and likely
identify specific model discrepancies. For example, in Figure 6a models A and E are alone in
identifying a population of parcels with low-HOOH that also have low-NOx. If this is not found
in the observations, then we have some clues (also looking at other key variables like in Figure
6b) that will identify locations and processes. Further, by looking at the reactivity of these
parcels (Figure 7), we can find that this region is important for methane and ozone loss. Some
work remains in establishing just how close is good enough in matching 2D (and multi-D)
probability distributions of the key species.

There are other ATom measurements beyond just key species that might prove useful as
climatological tests for the models. The OH loss frequency (L-OH, Sinha et al. (2008); Mao et
al. (2009)) is primarily determined by the longer-lived reactive species listed here, can be derived



from the key species, but it is not really a product of the 3D models. The models' predicted L-OH using their own key species could be tested with the L-OH observations, but then we are just testing the model's key species and our direct comparisons are more useful. Actinic fluxes and thus J-values are being measured by ATom and can be analyzed on a case-by-case basis

(Palancar et al., 2011) to assess the role of clouds in determining instantaneous reactivity. To be useful as a climatological test of the 24-hour modeled J-values, one would need to build up 3D cloud statistics from the measured J-values over the large range of solar zenith angles and altitudes that ATom samples. At present there is no clear path to use these to improve the climatologies of L-CH4, L-O3, and P-O3.


The first deployment of four, ATom-1, completed in August 2016, with ATom-2 flights to begin in January 2017. ATom was successful in completing all flights with instruments working, and acquiring well over 90% of the proposed data set, and measuring more than 30,000 10-second air parcels   A quick look at the pre-ATom planned flight tracks and sampling in Figure 9 shows the

coverage of the ocean basins, the large numbers of profiles, and the sampling frequency as a function of altitude. The expected release of ATom-1 data is mid-2017 and will include the global chemical model products discussed here. These measurements and analysis will provide a new approach for understanding which air matters.




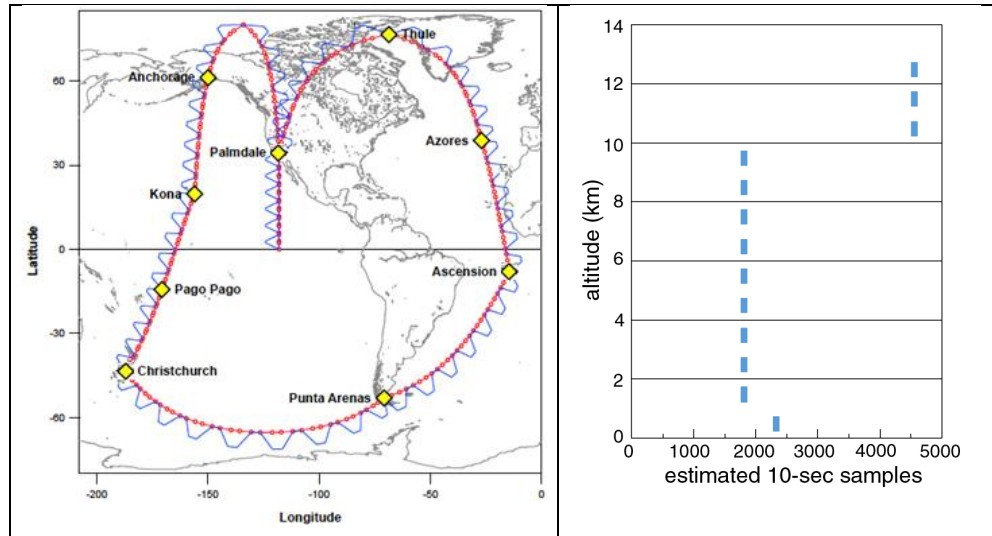

Figure 9. ATom proposed flight tracks (left) and estimated sampling frequency by 1-km altitude bins (right). The actual flights are somewhat altered. The altitude sampling is based on the proposed ~90 hours of flight time, ~180 profiles taking ~35 min for each pair of climb-descend, and 5 min spent each profile in the marine boundary layer. For up to date information on the ATom mission and deployments, see https://espo.nasa.gov/missions/atom/content/ATom.

***Acknowledgments***. This work was supported by NASA funding of the EVS2 Atmospheric Tomography (ATom) mission through a range of specific funding mechanisms.

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
