# Peer review of "Global Atmospheric Chemistry – Which Air Matters"

_Atmospheric Chemistry and Physics, 2016_

## Referee Comment (RC1) · Anonymous Referee #1 · 10 Feb 2017

General comments

-The manuscript presents an interesting method to test chemistry-climate models (CCMs) based on aircraft measurements. The method is motivated by the limited representativeness of aircraft measurements for the large grid cells of CCMs. While the proposed method can contribute to the evaluation of CCMs, alternative methods should be considered as well. For example, CCMs can assimilate meteorological analyses (model nudging) to realistically represent atmospheric conditions, so that the output can be directly compared to measurement data. And even though the grid size of models is course compared to aircraft measurements, some of the sub-grid scale variability can be accounted for by interpolating model results across the flight tracks.

-The manuscript includes chemistry-transport models (CTMs) for which it should be

unproblematic to represent realistic meteorology. Therefore, the method is particularly useful for CCM groups who have not implemented nudging techniques. Please mention this. Nevertheless, the new method is a practical and interesting addition to the techniques used to evaluate CCMs and CTMs. A more extensive discussion of pros and cons of different methods is recommended.

- The manuscript generally reads well, but the abstract and introduction need improvement. The introduction presents confusing statements, and the reader is guessing what the work actually focuses on up to L123. Please consider a more traditional structure of the introduction and present the concepts of representativeness (measurements vs model output) and chemical reactivity early on. I found the text partly confusing and unclear, often not to the point and sometimes not relevant. Examples and details are given below.

-Provided that the presentation, notably of the abstract and introduction, are improved, and the application of the method is described in the context of alternative methods, the manuscript will be acceptable for publication in ACP. However, since a methodology rather than new science is presented, it may also be considered to refer this manuscript to GMD, or add to the title that it is a technical note.

-Sect. 5 seems to suggest that the NASA A Tom missions are unique in providing high-frequency measurements across the Pacific and Atlantic Oceans. Such measurements have been made since decades, which is a great asset to the atmospheric chemistry community and in particular modelers (probably underused), and it should be mentioned that these datasets can also be applied in the presented methodology. The focus on the NASA A Tom missions in the manuscript are distinctive, which does not do justice to the more general applications of the method.

-The ending of the paper is unsatisfactory, with a promise of interesting things to come. Is it possible to present a sneak preview? This could make the manuscript more scientifically interesting rather than only technical, and help justify that it is published in ACP

rather than GMD.

Specific comments

The title is not very informative about the paper contents. Please reconsider.

L27: Few tenths of km?

L9-34 present technical details, before the reader knows what the work is about. Better first explain the objectives. And/or consider deleting these sentences.

L45: Understanding chemical heterogeneity is important, but not a sole prerequisite to understand global atmospheric chemistry. Please reformulate.

L48: Depends on where you are and what you are looking at. Please reformulate.

L52: "we ask which air is more important for the chemical evolution of human-driven air pollution". More important in view of what?

L55-56: What do you mean?

L57: Do you aim to understand the impact of heterogeneity?

L58: Thus, the integration of chemical reactivity controls the residence time of ozone? Please reformulate.

L67-68: see Chatfield and Delany, 1990.

L71: It may be helpful to define "heterogeneity" more precisely. Related to sub-grid scale processes? Dependence on grid size and compound considered? First you suggest that we simplify heterogeneity in view of modeling (L63-66) and now you use a model result to illustrate heterogeneity.

L76: Atmospheric rivers are narrower and related to tropical-extra-tropical interactions.

L78-79: Why/how does this indicate that chemical heterogeneity plays a role?

L79: Figure 1 seems trivial and could be omitted.

L84: Depends on scale.

L110-113: Nudging to meteorological analyses is typically applied to avoid this problem.

L114-119: Please also consider comparing CTM or nudged-CCM output at the location and time of the measurements. See S4D routine of Jöckel et al., 2010 (http://www.geosci-model-dev.net/3/717/2010/).

L123, L128: Now the central issue becomes clear: representativeness. It would be helpful to present this earlier in the abstract and introduction.

L189: Please explain why.

---

## Author Comment (AC1) · 17 Feb 2017

The authors thank referee #1 for the prompt, thoughtful, and constructive comments (RC1). Many of these are technical and will be corrected in the revision after all review comments are in (with fully traceable responses). Here, we think it is important to discuss some of the larger issues where our miswriting or omissions may have caused some misinterpretation.

C1) Alternative methods for model-measurement analysis using historical/nudged/specified dynamics in the models – Discussed already? We believe that lines 100-114 directly address this issue. Nudged CCMs or CCMs running with 'specified dynamics' are truly different models than the free-running climate versions. The nudging is an acceleration term that creates different residual tracer circulations

and even water cycles. If this 15-line explanation with references needs expanding, then we can.

C2) Methodology vs science paper – Both. Yes, there is a fair amount of nuts-and-bolts in this paper, and it needs to be documented, which might make it an AMT-like paper (GMD would be more for documenting specific models). Figures 1 and 2 are certainly motivation/methodology figures but beginning with Figure 3 we present new diagnostics that clearly identify how models differ. This paper presents a 6-model comparison of tropospheric reactivity using 2D and weighted probability distributions. The RC1 point is well made: we failed in the current manuscript to highlight the scientific conclusions from this comparison; thanks for catching this; we will fix it. In addition to the Conclusions, the Introduction (lines 146 & 167?) needs revision to point to what we found with the multi-model comparison.

C3) Other unbiased data sets – Yes, need to add. While we use some results from the incredibly valuable MOZAIC/IAGOS aircraft measurements and reference these papers, we failed to emphasize their importance in already providing an unbiased climatology for O3 and CO at cruise levels and for profiles over airports. Starting at line 136 we need to add this discussion. The major extension with ATom is the near-complete chemical package and the regular near global profiling. If the referee knows of other objective sampling chemical climatologies, please let us know and we will include in the revised discussion.
* * *

---

## Short Comment (SC1) · 13 Mar 2017

I am writing to note the submission by Prather et al., to examine probability distribution functions weighted by chemical reactivity, seems to be a promising complement to other approaches being developed to *quantify* differences in the representation of OH and CH4 lifetime within CCMs and CTMs.

The community eagerly awaits the data from ATom, and the approach outlined in the submitted paper will likely advance our understanding of not only why models differ in their treatment of OH and CH4 lifetime, but which models might actually be closer to the truth.

At the same time, I am sympathetic to the comment of the reviewer who stated:

-Sect. 5 seems to suggest that the NASA A Tom missions are unique in providing high-frequency measurements across the Pacific and Atlantic Oceans. Such measurements have been made since decades, which is a great asset to the atmospheric chemistry community and in particular modelers (probably underused), and it should be mentioned that these datasets can also be applied in the presented methodology. The focus on the NASA A Tom missions in the manuscript are distinctive, which does not do justice to the more general applications of the method.

For instance, the recently completed CONTRAST campaign is in the long-line of missions that have reported publicly available, high-frequency measurements in remote regions of the troposphere.

Should this paper proceed, Prather et al. might want to cite the usefulness of chemical fingerprinting via use of emission ratios as well as trajectory-based analyses for typing filaments to specific source regions, such as recently published by Anderson et al. (Nature Communications, 2016 http://www.nature.com/articles/ncomms10267) as another, complimentary means to look at this type of measurements.

Finally, the use of the CTM/CCMs in the A-run mode is a fascinating idea. Perhaps this will break the log-jam the community presently faces, driven by the difficulty in separating differences between OH precursors and chemical mechanism, with regards to model differences in OH and CH4 lifetime. At the same time should the editor decide that this paper will proceed, Prather et al. might want to note another new, recently published, promising approach: the use of neural networks (NNs) trained using archived, model output to simulated the chemical mechanism of each global model (i.e., Nicely et al., JGR, 2017 http://onlinelibrary.wiley.com/doi/10.1002/2016JD026239/full)

Personally, I hope this paper does proceed because I think the use of NNs (which require groups to archive specific quantities) versus special, new runs such as the A-run mode (which require groups to "disable processes that connect and mix air parcels" (line 318) will be a ripe discussion point among modelers at future meetings such as

the CCMI meeting being held 13-15 June 2017 in Toulouse, France.

---

## Referee Comment (RC2) · Anonymous Referee #2 · 27 Apr 2017

General comments The manuscript presents an interesting new approach for the comparison of observed atmospheric in-situ data alongside an aircraft flight track and results from chemistry-climate models. Such a comparison has the intrinsic problem that the observational data are related to specific conditions of time and geographical location which cannot be reproduced adequately within a global model due to their coarse grid and difficulties to represent real transport, emission, and deposition processes. This is related to the question of representativeness of the observations and the full variability of atmospheric constituents. The manuscript tries to add a new approach to overcome these problems by including more of the climatology of the model data and by shifting the comparison of single constituents towards the comparison of relationships between constituents including an additional weighting by processes like methane oxidation and ozone production. My feeling here is that both approaches (to-

gether or each of them separately) are very promising but the final concept how to do the comparison between observations and models is missing. Since the title, abstract, and introduction of the manuscript builds up a certain expectation about the presentation of a new and better tool for this kind of comparison I did feel quite disappointed at the end. The manuscript explains how to generate relationships between constituents and the additional weighting processes, both for the observations and for the model data, but the comparisons are then done by visual inspection and their subjective interpretation. Since I myself at some occasions would come to different interpretations looking at the presented figures of relationships between constituents, both weighted or unweighted, it is clear that this new approach is not facilitating the developement of objective arguments, at least at the moment.

Detailed comments: As I see it, the new approach in the manuscript contains three different parts. The first one shows how to compare between 3-D model results and observed data. The traditional way would be to compare observed data obtained at a specific time and location to model results from matching grid boxes for the same time of day and year. The new way is to compare model data from a broader region and time so that the exact match to the location and time of the observation is no intrinsic requirement any longer. This broadens the view because the model might be able to reproduce observed phenomena not at the exact coordinates but at a slightly different time or location. The key point here is: are the model climatology of a species and the unbiased climatology of observations similar or not. The second part demonstrates how to calculate important chemical reaction pathways with respect to methane loss and ozone loss/production (called reactivities) for a list of observed key species without the observation of reaction partners like HOx radicals. This is done by preparing a spin-up time of the involved CCM models, initializing the observed key species in the appropriate grid boxes of the model, switching-off model processes like transport, emission, deposition, and then propagation the single boxes in a 0-D fashion 24h into the future. The reactivities are then calculated as 24 hour averages of the processes methane oxidation, ozone production and destruction, and attributing these results to

the observed data. These reactivities are also calculated for the model data itself just by switching–off transport, emission, and deposition without initializing the key species to observed values. The manuscript tries to provide evidence that the switching-off of transport, emission, and deposition does not alter the reactivities too much by comparing full and switched-off model runs. But since the reactivities are 24h averages, the concentrations of the species are the same at the beginning of this 24h period shifting away only slowly afterwards, and since all species are in a kind of balance to each other due to the spin-up period, the real information content of this exercise might be less than anticipated. The exercise by initializing the key species to different values than the models own values is not done. Since such new values of the key species might not be in balance with the other species calculated during the model in the spin-up period, there might be discrepancies when the system tries to adjust to a new balance within the following 24h. The third part of the approach is to compare relationships between key species, either weighted or unweighted by the reactivities, rather than the key species themselves. I find this kind of approach very interesting since the underlying atmospheric processes introduce complex couplings between different species. Looking at a single species might give a wrong impression because good agreement between model and observation can be caused by compensating errors. But how to compare these relationships prepared from model data and from observations? There is no objective tool proposed to do that. The interpretation of a match or mismatch looking at relationships in the observations or in the model results still remains to be a subjective choice of the authors or readers of a manuscript. This gap is mentioned in the paper itself at line 645. Overall, I find that the proposed approach might have a lot of advantages, but its real capability is not proven or even is not illustrated. At the moment, the authors only can show results of climatologies of data calculated by six different CCMs. I think that one of them, model D, is quite different to the other five, for example looking at the probability distribution of HCHO in figure 5, the large values of J-O3(1D ) (together with model B) in figure 4, the large values of P-O3 at 20N-60N in figure 3, the large values of L-O3 at 20N-60N in figure 3, or the small values of L-

CH4 at 20S-20N in figure 3. If one would take the results of this model D as kind of observed dataset along the fight track shown in figure 9 and would follow the proposed approach, can one expect to reveal the difference of model D compared to the other five? This kind of question is already started at line 547 going to line 561. But besides mentioning some of the differences of model D there is no clue to explain its different behavior. If such an explanation is not possible for model D, can we really expect to learn something once observed data from ATom are available?
* * *

---

## Author Response (AR1)

**Overall Response – big issues**

Two big issues with this long and complex paper have been raised by the reviews: (1) what is the science here?, and (2) can you be more objective in comparing the probability distributions. Both points are valid. Thus, we returned to the paper to revise the conclusions to summarize what we learned from the multi-model comparison using our new diagnostics. Second we added a Supplement giving reduced/abbreviated statistics for the many plots in the main paper. We compared both standard deviation and equivalent percentiles, and developed a fitted ellipse approach to summarize the 2D plots. All of this has greatly improved the manuscript. These revisions took almost three weeks. We ask for a reasonably prompt response from the editor and reviewers, since this manuscript was submitted on December 7[th], was clearly lost in the holiday season, and thus has spent nearly 5 months in review, before any revisions.

**Response to Anonymous Referee #1**

> General comments. The manuscript presents an interesting method to test chemistry-climate models (CCMs) based on aircraft measurements. The method is motivated by the limited representativeness of aircraft measurements for the large grid cells of CCMs. While the proposed method can contribute to the evaluation of CCMs, alternative methods should be considered as well. For example, CCMs can assimilate meteorological analyses (model nudging) to realistically represent atmospheric conditions, so that the output can be directly compared to measurement data. And even though the grid size of models is course compared to aircraft measurements, some of the sub-grid scale variability can be accounted for by interpolating model results across the flight tracks.

The authors thank referee #1 for the prompt, thoughtful, and constructive comments (RC1). Many of these are technical and are corrected below as noted, but some are stylistic and the authors choose to retain their own style. This manuscript contains enough new material and approaches in developing chemical climatologies, and we cannot really evaluate too much more. The goal of a chemical climatology and its use for CCMs is already discussed and highlighted.

> -The manuscript includes chemistry-transport models (CTMs) for which it should be unproblematic to represent realistic meteorology. Therefore, the method is particularly useful for CCM groups who have not implemented nudging techniques. Please mention this. Nevertheless, the new method is a practical and interesting addition to the techniques used to evaluate CCMs and CTMs. A more extensive discussion of pros and cons of different methods is recommended.

Lines 100-114 directly address this issue. Nudged CCMs or CCMs running with 'specified dynamics' are truly different models than the free-running climate versions. The nudging is an acceleration term that creates different residual tracer circulations and even water cycles. We do not see a need to expand this 15-line explanation.

> -The manuscript generally reads well, but the abstract and introduction need improvement. The introduction presents confusing statements, and the reader is guessing what the work actually focuses on up to L123. Please consider a more traditional structure of the introduction and present the concepts of representativeness (measurements vs model output) and chemical reactivity early on. I found the text partly confusing and unclear, often not to the point and sometimes not relevant. Examples and details are given below.

Yes, the abstract and introduction have been edited to reflect many of the referee's detailed comments, and this greatly improves the presentation. There appears to be a stylistic issue with this referee and even a misunderstanding of the importance and timing of the discussion of representativeness. Representativeness comes later in the discussion, first we need to address heterogeneity and parcel sampling and how to build up statistics. Then we need to discuss reactivity weighting of the parcels. Figure 1 is needed to demonstrate the filamentary nature of reactivity. Representativeness applies first to the measurements to ensure that we have a reproducible climatology – see Figure 8 using models as an example for the sampling.

Overall we have gone through the introduction and tightened it based on the referee comments below and the apparent misunderstanding.

> -Provided that the presentation, notably of the abstract and introduction, are improved, and the application of the method is described in the context of alternative methods, the manuscript will be acceptable for publication in ACP. However, since a methodology rather than new science is presented, it may also be considered to refer this manuscript to GMD, or add to the title that it is a technical note.

We disagree with the referee's view of this manuscript: we believe it does provide a totally new model comparison across 6 major CCM/CTMs; and it does identify unique patterns of differences. In term of alternative methods, the referee is not clear, but we have added a Supplement that tabulates all the standard statistics for the plotted probability densities. We hope this helps. In addition, the introduction (lines 146 & 167) have been revised to point to what we found with the multi-model comparison.

> -Sect. 5 seems to suggest that the NASA ATom missions are unique in providing high-frequency measurements across the Pacific and Atlantic Oceans. Such measurements have been made since decades, which is a great asset to the atmospheric chemistry community and in particular modelers (probably underused), and it should be mentioned that these datasets can also be applied in the presented methodology. The focus on the NASA ATom missions in the manuscript are distinctive, which does not do justice to the more general applications of the method.

While we note some results from the incredibly valuable MOZAIC/IAGOS aircraft measurements and reference these papers, we failed to emphasize their importance in providing an unbiased climatology for $O_3$ and CO at cruise levels and for profiles over airports. We also refer to the early exploratory missions that were also fairly objective searches of atmospheric composition. Starting at line 136 we have augmented the Introduction to add this discussion. The major extension with ATom is the near-complete chemical package and the regular near

global profiling.  If the referee knows of other objective sampling chemical climatologies, please let us know and we will include in the revised discussion.

> -The ending of the paper is unsatisfactory, with a promise of interesting things to come. Is it possible to present a sneak preview? This could make the manuscript more scientifically interesting rather than only technical, and help justify that it is published in ACP.

On this we agree with the referee and have strengthened the conclusion section, including new information and findings about the tropospheric chemistry of the 6 models, while leaving in a mention of the opportunities with Atom.

> Specific comments. The title is not very informative about the paper contents. Please reconsider.

We reconsidered, and believe the title is sound and accurately describes what is new here. There are many possible titles one could come up with, and this is a stylistic choice of the authors.

> L27: Few tenths of km?

Thanks, done.

> L29-34 present technical details, before the reader knows what the work is about. Better first explain the objectives. And/or consider deleting these sentences.

Good point. We have inserted a phrase at Line 29 to explain the objectives of the research.  The other sentences are needed to flesh out the objectives.

> L45: Understanding chemical heterogeneity is important, but not a sole prerequisite to understand global atmospheric chemistry. Please reformulate.

The text as is makes only the point that heterogeneity is one aspect that must be understood, if one wants to understand global tropospheric chemistry.  We did not say it, and it is certainly NOT the sole issue, we agree there.  No change is needed, as style, putting this first is the authors' choice.

> L48: Depends on where you are and what you are looking at. Please reformulate.

This statement is basically correct – it refers to the chemical mix from the previous sentence.  It is hard to imagine a place with truly isolated air parcels for long periods of time:  there are always gradients at some levels and mixing across shear zones.  The only example of quasi-seasonal isolation occurs in the summer stratosphere, and that does not last through the year.  This seems to be a stylistic issue, not changed.

> L52: "we ask which air is more important for the chemical evolution of human-driven air pollution". More important in view of what?

Thanks, revised to explicitly say chemical evolution of the global tropospheric pollution. (instead of pollution in general). Methane and ozone are introduced later, they do not fit here.

> L55-56: What do you mean?

We believe that the sentence can be understood simply as is: If we know the chemical mix of all the air parcels, then we can use fingerprinting or trajectories or other methods to identify the level of human influence. Have added "and where they come from".

> L57: Do you aim to understand the impact of heterogeneity?

Yes, we do. We will look at how the high/low ends of the statistical distribution affect the net rates. For example are they more important than their frequency of occurrence. This is a non-linear system, especially when one considers the co-variance of other key species. Can we just average all he species and get the same rates?

> L58: Thus, the integration of chemical reactivity controls the residence time of ozone? Please reformulate.

Yes, this was cryptic. We have added "atmosphere's" to describe the integration. It is not our integration but the atmosphere's integration. We also changed the "residence time" to "evolution".

> L67-68: see Chatfield and Delany, 1990.

Thanks, yes, a good choice. We have added the reference to the non-linear discussion here.

> L71: It may be helpful to define "heterogeneity" more precisely. Related to sub-grid scale processes? Dependence on grid size and compound considered? First you suggest that we simplify heterogeneity in view of modeling (L63-66) and now you use a model result to illustrate heterogeneity.

We think this is clear. We do NOT suggest that we simplify heterogeneity, but rather than many modeling efforts do so.

> L76: Atmospheric rivers are narrower and related to tropical-extra-tropical interactions.

Atmospheric rivers in a column sense may not be narrower. The text has been edited to clarify that atmos-rivers are not LCH4-rivers. And indeed , ti makes it even more interesting to discover what causes the LCH4-rivers.

> L78-79: Why/how does this indicate that chemical heterogeneity plays a role?

Obviously the chemical composition across the rivers of LCH4 is not homogeneous. Clearly there are some interesting heterogeneous patterns out there, at the reactivity level.

> L79: Figure 1 seems trivial and could be omitted.

Sorry Figure 1 is distinct and shows an aspect of atmospheric chemistry that remains largely unnoted and unpublished. While atmospheric rivers of water have many references, we could not find any to these rivers of chemical reactivity.

> L84: Depends on scale.

Thanks, yes, have replaced 'ways' with "patterns" which includes the space-time scales of emissions.

> L110-113: Nudging to meteorological analyses is typically applied to avoid this problem.

> L114-119: Please also consider comparing CTM or nudged-CCM output at the location and time of the measurements. See S4D routine of Jöckel et al., 2010 (http://www.geosci-model-dev.net/3/717/2010/).

See earlier comments. A free-running and a nudged CCM are entirely different models. Convection, clouds and everything changes. There are usually hidden residual transports with such nudging, sicne one has to correct the model's trends with winds and rain to the observations. The reference Joeckel et al work is one way to assimilate met data, but that makes it a different model. Moreover, the models that nudge/assimilate/forecast meteorology never seem to agree anyway.

> L123, L128: Now the central issue becomes clear: representativeness. It would be helpful to present this earlier in the abstract and introduction.

See earlier comments. Representativeness is but one of the problems. If there is sufficient sampling (e.g., from a model) then R is not a problem. With limited observations we must question R. All the ground work must be laid before we discuss R.

> L189: Please explain why.

Yes, good point. We have added the explanatory sentence at line 191. " L-CH4 is highly variable across parcels, and the integral of L-CH4 determines the atmospheric lifetime of $CH_4$ and the buildup of its emissions in the atmosphere."

**Response to Anonymous Referee #2**

> *General comments The manuscript presents an interesting new approach for the comparison of observed atmospheric in-situ data alongside an aircraft flight track and results from chemistry-climate models. Such a comparison has the intrinsic problem that the observational data are related to specific conditions of time and geographical location which cannot be reproduced adequately within a global model due to their coarse grid and difficulties to represent real transport, emission, and deposition processes. This is related to the question of representativeness of the observations and the full variability of atmospheric constituents.*

We concur and these aspects are discussed in the paper.

> *The manuscript tries to add a new approach to overcome these problems by including more of the climatology of the model data and by shifting the comparison of single constituents towards the comparison of relationships between constituents including an additional weighting by processes like methane oxidation and ozone production. My feeling here is that both approaches (together or each of them separately) are very promising but the final concept how to do the comparison between observations and models is missing.*

We agree in part, but as the reviewer notes, we have begun pushing new types of promising comparisons that may yet prove fruitful. Even with my best optimism, I could never presume that we could come up with a 'final concept.'

Nevertheless, the point is taken and we have spent considerable time developing a large Supplement to the manuscript that summarizes the plots in simpler statistics (percentiles, means, standard deviations, even 2D fitted ellipses), which allow for more ready comparisons. The model comparisons and case studies shown have not changed and thus the basic conclusions remain, but the more objective statistics as suggested by the referee have made the distinctions between models more objective. Thanks. Changes to the text occur throughout as we link to the Supplement, and Fig 8 has added a plot of the fitted ellipses to the transect examples. See further discussion in next comment.

> *Since the title, abstract, and introduction of the manuscript builds up a certain expectation about the presentation of a new and better tool for this kind of comparison I did feel quite disappointed at the end. The manuscript explains how to generate relationships between constituents and the additional weighting processes, both for the observations and for the model data, but the comparisons are then done by visual inspection and their subjective interpretation. Since I myself at some occasions would come to different interpretations looking at the presented figures of relationships between constituents, both weighted or unweighted, it is clear that this new approach is not facilitating the development of objective arguments, at least at the moment.*

This comment is true and the community has been wrestling with comparing complex structures of chemical constituents (e.g., color contours) and then decreeing that "B looks pretty much like A" or the counterpart. We did not have an easy go-to method, but took on the task of converting the figures into fewer statistics in the Supplement described above. These are now introduced in the paper and for most figures and at least represent a first step. For 1D PDs we tabulate both the percentiles (16%-50%-84%) and compare with the mean plus or minus one standard deviation. All of these statistics are calculated in $\log_{10}([X])$ where $[X]$ is the mole fraction in ppt. The bin width is set to 0.05. This is the correct choice as we find most PDs to be log normal in that the percentiles match the standard deviation in $\log_{10}$ space. For 2D PDs we could not easily find a percentile approach and so stick to the Gaussian statistics. The diagnostics here are a bit more exotic: the mean centroid and a single 2D combined variance (sigma-x^2 + sigma-y^2) are obvious; next we rotate the data about the centroid (conserves the 2D variance) until the ratio sigma-y/sigma-x is at a minimum (i.e., the flattest ellipse); and we record the angle of rotation and the minimum ratio. This now gives us just 5 numbers to describe the 2D PDs; it also allows us to clearly quantify tight quasi-linear relationships; and it opens up methods to calculate an overlap fraction of the ellipses between two PDs. A new didactic figure of how this works is supplied as Fig S1.

> *Detailed comments: As I see it, the new approach in the manuscript contains three different parts. The first one shows how to compare between 3-D model results and observed data. The traditional way would be to compare observed data obtained at a specific time and location to model results from matching grid boxes for the same time of*

*day and year. The new way is to compare model data from a broader region and time so that the exact match to the location and time of the observation is no intrinsic requirement any longer. This broadens the view because the model might be able to reproduce observed phenomena not at the exact coordinates but at a slightly different time or location. The key point here is: are the model climatology of a species and the unbiased climatology of observations similar or not. The second part demonstrates how to calculate important chemical reaction pathways with respect to methane loss and ozone loss/production (called reactivities) for a list of observed key species without the observation of reaction partners like HOx radicals. This is done by preparing a spin-up time of the involved CCM models, initializing the observed key species in the appropriate grid boxes of the model, switching-off model processes like transport, emission, deposition, and then propagation the single boxes in a 0-D fashion 24h into the future. The reactivities are then calculated as 24 hour averages of the processes methane oxidation, ozone production and destruction, and attributing these results to transport, emission, and deposition without initializing the key species to observed values.*

Yes, this was our intent. We have already achieved a major threshold in this paper with 6 major chemistry models having implemented the A-run format successfully that will allow them to ingest an observational data stream of key species when they become available (e.g., ATom-1 will become public in the latter part of this year). Such measurements should be able to establish constraints on large-scale "reactivities" as calculated by different models, and thus the agreement/spread of model results with measurements can place some constraints on model predictions for future atmospheres.

*The manuscript tries to provide evidence that the switching-off of transport, emission, and deposition does not alter the reactivities too much by comparing full and switched-off model runs. But since the reactivities are 24h averages, the concentrations of the species are the same at the beginning of this 24h period shifting away only slowly afterwards, and since all species are in a kind of balance to each other due to the spin-up period, the real information content of this exercise might be less than anticipated. The exercise by initializing the key species to different values than the models own values is not done. Since such new values of the key species might not be in balance with the other species calculated during the model in the spin-up period, there might be discrepancies when the system tries to adjust to a new balance within the following 24h.*

As the reviewer will probably agree, there is enough new material in this paper that needs to be examined, played with, and understood. The final point above is already underway as we have distributed a simulated observing transect from an independent model. Half of the models have completed this exercise and it will be a short, GRL-length paper this summer. We await the public release of the ATom data when the quality controls and calibrations are complete. This is promised by July, and that will begin a new stage of investigation along these lines.

*The third part of the approach is to compare relationships between key species, either weighted or unweighted by the reactivities, rather than the key species themselves. I find this kind of approach very interesting since the underlying atmospheric processes introduce complex couplings between different species. Looking at a single species might give a wrong impression because good agreement between model and observation can be caused by compensating errors. But how to compare these*

*relationships prepared from model data and from observations? There is no objective tool proposed to do that. The interpretation of a match or mismatch looking at relationships in the observations or in the model results still remains to be a subjective choice of the authors or readers of a manuscript. This gap is mentioned in the paper itself at line 645.*

Yes, we have introduced quantitative values for the PDs that allow one to address overlap of the weighted parcels in both 1D and 2D. We cannot full flesh this out here (the paper is gaining weight rapidly), but we expect to do this with the upcoming simulated observations paper and then the first paper using all of the ATom data stream in all the models.

*Overall, I find that the proposed approach might have a lot of advantages, but its real capability is not proven or even is not illustrated.*

The new brief statistics describing the distributions should help with illustration (see Fig S1), but the real capability of these new approaches is just beginning. I believe that we are still in exploration mode.

*At the moment, the authors only can show results of climatologies of data calculated by six different CCMs. I think that one of them, model D, is quite different to the other five, for example looking at the probability distribution of HCHO in figure 5, the large values of J-O3(1D ) (together with model B) in figure 4, the large values of P-O3 at 20N-60N in figure 3, the large values of L-O3 at 20N-60N in figure 3, or the small values of L-CH4 at 20S-20N in figure 3. If one would take the results of this model D as kind of observed dataset along the fight track shown in figure 9 and would follow the proposed approach, can one expect to reveal the difference of model D compared to the other five? This kind of question is already started at line 547 going to line 561.*

Yes, and with the next (pre-ATom release) paper we will see if model D's differences are from the initial species as predicted by model D or from its photochemistry module. From the 1D statistics alone, it is clear that model D has a very different distribution of reactivity since the weighted means shift differently than the other models (Table S5).

*But besides mentioning some of the differences of model D there is no clue to explain its different behavior. If such an explanation is not possible for model D, can we really expect to learn something once observed data from ATom are available?*

We thank this reviewer for a careful read about what is new in this manuscript and what is missing. We have added a major new component (the statistics supplement) but cannot expand to include all of the useful suggestions. Identifying major differences in model results is a first-order question for the community. This manuscript starts this process with the comparison of six global models. This reviewer clearly understands the complexity of current 3D chemistry models, and must realize how hard it is to identify a cause of the model differences. This thankless task is really up to the originator and developer of that model who can understand the inner tickings. What we provide here are valuable diagnostics (esp. with paper #2) to allow more educated tracing of model differences (whether against other models or observations, and when constrained by initialization of key species).

**_Response to R. J. Salawitch Comments_**

> _I am writing to note the submission by Prather et al., to examine probability distribution functions weighted by chemical reactivity, seems to be a promising complement to other approaches being developed to *quantify* differences in the representation of OH and CH4 lifetime within CCMs and CTMs._

Thanks for the encouragement.

> _The community eagerly awaits the data from ATom, and the approach outlined in the submitted paper will likely advance our understanding of not only why models differ in their treatment of OH and CH4 lifetime, but which models might actually be closer to the truth._

Yes, the models here are also awaiting the first archival release of ATom data later this summer.

> _At the same time, I am sympathetic to the comment of the reviewer who stated:_

> _Sect. 5 seems to suggest that the NASA A Tom missions are unique in providing high-frequency measurements across the Pacific and Atlantic Oceans. Such measurements have been made since decades, which is a great asset to the atmospheric chemistry community and in particular modelers (probably underused), and it should be mentioned that these datasets can also be applied in the presented methodology. The focus on the NASA ATom missions in the manuscript are distinctive, which does not do justice to the more general applications of the method._

The original contributions and continued value of the other chemistry missions is very important, it is noted up front in the manuscript. ATom does have unique features that we highlight in Section 5. We believe it is the only such transect like mission with a chemical package that includes all the key species for reactivity and regular profiling from 60N to 60S. ATom avoids the subjective sampling which prevents us from using their data without a full hindcast-matchng with the chemical models (not possible with climate models).

> _For instance, the recently completed CONTRAST campaign is in the long-line of missions that have reported publicly available, high-frequency measurements in remote regions of the troposphere._

CONTRAST was a great mission, but it was admittedly a "process study" for measuring convection from the marine boundary layer to the UT/LS. As such it spent more time in the boundary layer and upper troposphere, rather than the middle troposphere (0.5 – 8 km) where most of the chemical reactivity occurs. With a well thought out adjustment of the sampling patterns, it could be usefully merged with the ATom data. A reference to the CONTRAST mission and to the box modeling is now in the manuscript.

> _Should this paper proceed, Prather et al. might want to cite the usefulness of chemical fingerprinting via use of emission ratios as well as trajectory-based analyses for typing filaments to specific source regions, such as recently published by Anderson et al._

> *(Nature Communications, 2016 http://www.nature.com/articles/ncomms10267) as*
> *another, complimentary means to look at this type of measurements.*

The work here is to identify which parcels are highly reactive and controlling the O3 and CH4 abundances. After these have been identified along with the frequency of their occurrence, then someone else can work on attributing sources for these hot spots.

> *Finally, the use of the CTM/CCMs in the A-run mode is a fascinating idea. Perhaps this*
> *will break the log-jam the community presently faces, driven by the difficulty in sepa-*
> *rating differences between OH precursors and chemical mechanism, with regards to*
> *model differences in OH and CH4 lifetime. At the same time should the editor decide*
> *that this paper will proceed, Prather et al. might want to note another new, recently*
> *published, promising approach: the use of neural networks (NNs) trained using archived,*
> *model output to simulated the chemical mechanism of each global model (i.e., Nicely et*
> *al., JGR, 2017 http://onlinelibrary.wiley.com/doi/10.1002/2016JD026239/full)*

We acknowledge that neural networks can provide a very useful abstraction of the complex and certainly obscure workings of these CCMs. Indeed, the dataset collected from the six models for ATom could provide a sample of how the models calculate O3 and CH4 production and loss. So, the NNs provide a transfer standard for the chemical model but not for the resulting distribution of key species, whether this is better than running the models themselves, even in A-run mode as demonstrated here is an open question. We put in a reference to this paper in the A-run discussion (lines ~ 450ff).

> *Personally, I hope this paper does proceed because I think the use of NNs (which*
> *require groups to archive specific quantities) versus special, new runs such as the A- run*
> *mode (which require groups to "disable processes that connect and mix air parcels" (line*
> *318) will be a ripe discussion point among modelers at future meetings such as the*
> *CCMI meeting being held 13-15 June 2017 in Toulouse, France.*

Yes, it could.

[revised manuscript text omitted]

This Supplement examines some simple statistics that may be useful in providing a quantitative synthesis of the full probability distributions (shown in the main paper) so that different models or weightings can be compared and judged to be similar or be given a measure of their difference.  As in the main paper, the case here focusses on the tropical Pacific block (150E-

25   210E, 20S-20N, 0-12 km) for one day in the middle of August.  The model key is: A = CAM4-Chem; B = GEOS-Chem; C = GFDL-AM3; D = GISS-E2; E = GMI-CTM; F = UCI-CTM. The reactivities are calculated from the C-runs (standard CCM/CTM simulation).  The table numbers in the supplement are keyed to the figure numbers in the main paper and thus are not sequential, but the supplement figures are numbered sequentially.

**1D Probabilities**.  The distributions here are evaluated on log of abundance, expressed as mole fraction.  The bin spacing is in units of 0.05 for $\log_{10}$ for all probability densities.  In the right-hand columns of Tables S6a and S6b, we calculate the 1D probability statistics for the Pacific block for all 6 models for the species [HOOH], {NOx}, [$H_2O$], and [$O_3$].  The mean values

35   minus and plus 1 standard deviation are given in the upper part of each model-species cell; those of the 16th-50th-84th percentiles are given immediately underneath to facilitate comparison. In Table S8a we give similar statistics comparing the pacific block with single transects, sampling along single longitudes within the block (150E, 165E, 180E, 195E, 210E).  In this case we only show results for 2 models (C & F) and 2 species (HOOH & NOx).  Note that results for

40   the block itself in Table S8 are the same as in Table S6a.

For [HOOH], [NOx], and [$O_3$] the Gaussian statistics and the percentiles are almost identical, and thus the central 68% of the species distribution can be readily characterized.  For [$H_2O$] the extremely large spread in abundance across 0-12 km altitude range results in the median being consistently larger than the mean by about 0.2 and the percentile spread (16th to 84th %ile) being noticeably narrower than ± σ.

The 1D statistics are useful, for example, in demonstrating that model D has about 2x higher HOOH than other models and this difference applies across the ±σ range.  For NOx, all models have similar averages, but models C & F have almost 2x higher NOx at the 16th percentile. Five models have a similar value for the 84th %ile of NOx (1.5 = ~30 ppt), but model E is much

50   lower (~20 ppt).  Surprisingly, model A has notably higher $O_3$ across the range:  a 50th %ile value of ~45 ppt versus ~25 ppt for the other models.  Among the other five models, D and E

have a much lower 16th %ile $O_3$ (~13 ppt vs ~17 ppt).   Water vapor ($H_2O$) across the 0-12 km altitude range is expected to have a very large spread and indeed the 16th – 84th %ile spread ranges from 40x to 80x.  Model C is a climate model with prognostic $H_2O$ based on a full convection scheme. It has 2x less $H_2O$ than the other models and a much narrower ±σ range matching its 16th – 84th %ile range.

A more complete list of the 1D probability density statistics, based on Figure 5, is presented in Tables S5ab.  Six major gases are listed (HOOH, NOx, $H_2O$, $O_3$, CO, HCHO), and the effect of reactivity weighting is clearly seen.  CO has a very narrow distribution and the mean does not change with reactivity weighting.  What is clear here and better quantified is that model A's average CO is about 0.12 ($\log_{10}$) less than the other 5 models. For HOOH the pattern provides clues to the model differences.  For models ABCEF, the Air and PO3 weighting show similar statistics, but the LO3 and LCH4 weighting show higher average HOOH by about 0.10.  This reflects the greater weighting of the lower troposphere (with greater HOOH) in LO3 & LCH4. As noted above in discussing Fig 6, model D has much larger HOOH, but it also has different shifts across the 4 weighting factors.  For $H_2O$ all models show that the average with PO3 drops about 0.2-0.3 (upper troposphere dominates PO3) and that it increases by about 0.3-0.5 with LO3 & LCH4 weighting.  All models except D have similarly weighted averages for LO3 & LCH4, and this points to a dramatically different pattern for these two reactivities in model D.  For HCHO, models ABCEF have similar patterns with an average and standard deviation of about 2.2 and 0.3 for both Air & PO3 weighting.  This average increase to 2.3-2.4 with LO3 & LCH4 weighting but the standard deviation decreases to 0.2.  Model D again stands out with a different response to the weightings and a lower HCHO abundance by 0.3 (factor of 2).  For $O_3$, all models show a similar and small shift of the average with the different weightings, and model C is clearly and uniformly across ±σ higher than all other models, as noted in Fig. 6 discussion.  NOx has highest ±σ spread (other than H2O) of about 1.0 (factor of 10 in mole fraction).  All models agree that PO3 occurs in parcels with much higher NOx (by 0.13 to 0.31) and that LO3 & LCH4 occurs in parcels with lower NOx.  Model D again shows that LO3 and LCH4 occur in different parcels.  In comparing models C and F, one finds that both have similar amounts of NOx in Air & LO3 & LCH4 weighted parcels, but for PO3 model F has clearly higher NOx by 0.1 across the ±σ range.

**2D Probabilities**.  The colored plots of 2D probability density are harder to characterize simply. There are a range of multivariate statistics methods (e.g., Hotelling $T^2$ test) that can test whether two different modeled 2D density plots are demonstrably different, but it is easy to see that if we have enough samples, then almost all of the plots here are statistically different.  We need to gather some simple statistics from our 2D plots that are similar to the Gaussian statistics or percentiles of the 1D distributions.

We characterize the 2D densities with a fitted ellipse centered at the mean value ($X_0$, $Y_0$) with semi-major and semi-minor axes defined as the standard deviation in the two orthogonal axes. These axes are rotated until we find an optimum fit with flattest ellipse (i.e., the lowest ratio of semi-minor to semi-major axis).  The long axis of the ellipse is defined by a rotation angle counterclockwise from the X axis. Figure S1 shows an example of how such an ellipse appears on top of the 2D colored density distribution (from Fig 6a model B). Generally, about 40-50% of the weighted parcels fall within this ellipse, which is as expected for two independent normal distributions (0.68 x 0.68).  The 2D variance ($\sigma_X^2 + \sigma_Y^2$), calculated with respect to any (X,Y)-

axis located at the mean of the 2D probability distribution, is unchanged by rotation of the axes about the centroid and is defined here as $\sigma_{2D}$. All of these quantities, including the rotation angle, are given for the 6 models and the Pacific block in Tables S6a ((X, Y) = (HOOH, NOx)) and S6b ((X, Y) = ($H_2O$, $O_3$)).

Figures S2ab show the fitted ellipses for the 2D densities plotted in Figures 6ab. For (HOOH, NOx), models ABE have similar overlapping ellipses; whereas models CF have distinctly smaller but similarly angled and proportioned ellipses. Model D is distinct in terms of offset to higher HOOH values. For ($H_2O$, $O_3$), all models have similarly shaped and angled ellipses with rotation angles ranging only from 166 to 173 degrees. The difference in ellipse overlap is characterized simply by Y0, with models BCF having Y0 = 4.45±0.01, models DE having Y0 = 4.35±0.01, the previously identified outlier model A having Y0 = 4.64.

Tables 6cd give the area of each ellipse and the %-overlap with the other ellipses. We define overlap here to be symmetric (i.e., the overlapping area divided by the average of the two ellipse areas) so that a smaller ellipse fitting wholly within a larger ellipse does not result in 100% overlap. For (HOOH, NOx) the %-overlap shows clearly what we diagnosed subjectively from Figure S2a: the 3 %-overlap pairs for ABE and the one %-overlap for CF are all about 80%, much higher than for any other pairs, except for the self-overlap of 100%. The average %-overlap against the other 5 models, clearly identifies the outlier as model D. The question of whether the outlier is the more accurate model will await a similar comparison with observations. For ($H_2O$, $O_3$), this diagnostic shows that model A is the clear outlier and identifies the high-overlap pairs seen in the figure.

Figure 8 in the main paper looks at the fundamental uncertainty of representativeness by comparing the 2D densities for (HOOH, NOx) from the over-sampled Pacific block with the smaller samples along five longitudinal transects within the block (150E, 165E, 180E, 195E, 210E). As clearly seen in Figures 8ab for models CF, the single transect sampling is sparser and it is harder to see a clear pattern. The fitted ellipses to these transects in Figures 8cd are able to statistically characterize the distributions and show that the single transects accurately measure the block and that they also clearly distinguish the two models C and F as being similar, but different. The statistics from the Figure 8ab plots are given in Tables S8ab. The %-overlap is a compelling diagnostic: the one transect at 150E is clearly different from all others, and the other 4 transects overlap with modeled block ellipse at the 86% to 94% level. The 150E ellipse is clearly different when plotted (Fig. 8cd) and the %-overlap is only 40-60%, the same overlap between the two models. The 150E transect has the same longitude as Sydney and cuts through Papua New Guinea. This region is directly influenced by deep convection, lightning, and other continental sources, whereas the 165E-210E region looks similar, remote, and presumably representative of much of the Pacific basin. We conclude that fitted ellipses for comparing 2D probability densities is a valuable tool. Perhaps 3D+ distributions could be fitted in a similar manner to ellipsoids for analysis.

[Figure]

Figure S1. Model B (GEOS-Chem) air-mass weighted tropical Pacific distribution of parcels for the NOx vs. HOOH abundance showing the fitted ellipse. The species abundance is sampled in $\log_{10}$ space with a pixel resolution of 0.05 x 0.05. The color density pixels are the same as for model B in Figure 6a. For summary statistics on this distribution see Table S6a.

[Figure]

Figure S2. Fitted ellipses to the modeled 2D density distributions in (a) Figure 6a (HOOH, NOx) and (b) Figure 6b ($H_2O$, $O_3$). The data defining each ellipse and their %-overlap are given in Table S6ab.

Table S6a.  Indicators derived from the 2D distribution of (HOOH, NOx) shown in Fig 6a. All values are based on units of $\log_{10}([X, ppt])$.

| model | centroid $(X_0,Y_0)$ | $\sigma_{2D}$ | axes (major,minor) | rotation angle | HOOH X: -σ avg +σ X: 16-50-84 % | NOx Y: -σ avg +σ Y: 16-50-84 % |
|---|---|---|---|---|---|---|
| A | 2.60, 1.00 | 0.665 | 0.559, 0.361 | 98 | 2.24 2.60 2.97
2.22 2.61 2.95 | 0.44 1.00 1.55
0.35 1.04 1.51 |
| B | 2.63, 0.98 | 0.603 | 0.560, 0.223 | 109 | 2.35 2.63 2.92
2.37 2.66 2.86 | 0.45 0.98 1.51
0.35 1.01 1.48 |
| C | 2.75, 1.09 | 0.507 | 0.441, 0.251 | 134 | 2.39 2.75 3.10
2.36 2.82 3.06 | 0.73 1.09 1.46
0.68 1.07 1.44 |
| D | 2.96, 0.93 | 0.682 | 0.603, 0.318 | 119 | 2.56 2.96 3.36
2.60 2.98 3.30 | 0.38 0.93 1.48
0.37 0.87 1.48 |
| E | 2.59, 0.92 | 0.550 | 0.448, 0.319 | 94 | 2.27 2.59 2.91
2.26 2.58 2.89 | 0.47 0.92 1.37
0.40 0.92 1.34 |
| F | 2.70, 1.12 | 0.491 | 0.451, 0.195 | 115 | 2.44 2.70 2.96
2.40 2.72 2.92 | 0.70 1.12 1.54
0.62 1.16 1.51 |

Table S6b.  Indicators derived from the 2D distribution of (H2O, O3) shown in Fig 6b. All values are based on units of $\log_{10}([X, ppt])$.

| model | centroid $(X_0,Y_0)$ | $\sigma_{2D}$ | axes (major,minor) | rotation angle | $H_2O$ X: -σ avg +σ X: 16-50-84 % | $O_3$ Y: -σ avg +σ Y: 16-50-84 % |
|---|---|---|---|---|---|---|
| A | 3.49, 4.64 | 0.799 | 0.775, 0.149 | 167 | 2.74 3.49 4.25
2.58 3.65 4.27 | 4.42 4.65 4.87
4.36 4.67 4.85 |
| B | 3.46, 4.46 | 0.824 | 0.808, 0.162 | 167 | 2.67 3.46 4.25
2.50 3.62 4.26 | 4.22 4.46 4.71
4.18 4.47 4.69 |
| C | 3.30, 4.44 | 0.839 | 0.817, 0.144 | 173 | 2.49 3.30 4.11
2.34 3.35 4.21 | 4.27 4.44 4.62
4.24 4.43 4.59 |
| D | 3.59, 4.35 | 0.780 | 0.761, 0.171 | 166 | 2.85 3.59 4.33
2.74 3.77 4.29 | 4.11 4.35 4.60
4.07 4.34 4.60 |
| E | 3.58, 4.36 | 0.803 | 0.778, 0.200 | 169 | 2.81 3.58 4.34
2.68 3.82 4.27 | 4.12 4.36 4.61
4.06 4.34 4.60 |
| F | 3.42, 4.44 | 0.858 | 0.835, 0.149 | 171 | 2.59 3.42 4.24
2.39 3.60 4.27 | 4.24 4.44 4.64
4.21 4.43 4.62 |

Table 6c. Overlap percent of SD2 ellipse fit (NOx vs HOOH).

| model | area | x A | x B | x C | x D | x E | x F | x others |
|-------|-------|------|------|------|------|------|------|----------|
| A | 0.633 | 100% | | | | | | 64% |
| B | 0.387 | 74% | 100% | | | | | 61% |
| C | 0.346 | 59% | 58% | 100% | | | | 62% |
| D | 0.603 | 42% | 35% | 58% | 100% | | | 43% |
| E | 0.452 | 83% | 77% | 53% | 34% | 100% | | 60% |
| F | 0.283 | 62% | 62% | 81% | 48% | 55% | 100% | 62% |

Overlap % is defined as the overlapping area divided by the average of the two ellipse areas and is thus symmetric. The last column (x others) is the average overlap % against the other 5 models.

145

Table 6d. Overlap percent of SD2 ellipse fit ($O_3$ vs $H_2O$).

| model | area | x A | x B | x C | x D | x E | x F | x others |
|-------|-------|------|------|------|------|------|------|----------|
| A | 0.368 | 100% | | | | | | 18% |
| B | 0.407 | 30% | 100% | | | | | 65% |
| C | 0.361 | 13% | 72% | 100% | | | | 62% |
| D | 0.406 | 9% | 68% | 70% | 100% | | | 62% |
| E | 0.490 | 16% | 72% | 70% | 89% | 100% | | 65% |
| F | 0.396 | 20% | 86% | 86% | 71% | 77% | 100% | 68% |

150

| Model longitude | centroid $(X_0,Y_0)$ | $\sigma_{2D}$ | axes (major,minor) | rotation angle | HOOH X: -σ avg +σ / X: 16-50-84 % | NOx Y: -σ avg +σ / Y: 16-50-84 % |
|---|---|---|---|---|---|---|
| | | | | | | |

Table S8a. Indicators derived from the 2D distribution of (HOOH, NOx) as shown in Fig 8, including the SD2 ellipse. All values are based on units of $\log_{10}([X, ppt])$. The Block equals all grid cells 150E-201E, 20S-20N, 0-12 km.  Individual longitudes include only a single transect sampled.

| Model longitude | centroid $(X_0,Y_0)$ | $\sigma_{2D}$ | axes (major,minor) | rotation angle | HOOH X: -σ avg +σ  /  X: 16-50-84 % | NOx Y: -σ avg +σ  /  Y: 16-50-84 % |
|---|---|---|---|---|---|---|
| C Block | 2.75, 1.09 | 0.507 | 0.441, 0.251 | 134 | 2.39 2.75 3.10 / 2.36 2.82 3.06 | 0.73 1.09 1.46 / 0.68 1.07 1.44 |
| C 150E | 2.78, 1.27 | 0.526 | 0.414, 0.324 | 146 | 2.40 2.78 3.17 / 2.43 2.86 3.09 | 0.91 1.27 1.62 / 0.89 1.16 1.61 |
| C 165E | 2.77, 1.06 | 0.497 | 0.454, 0.203 | 130 | 2.44 2.77 3.10 / 2.42 2.81 3.08 | 0.69 1.07 1.44 / 0.65 1.01 1.42 |
| C 180E | 2.72, 1.10 | 0.502 | 0.436, 0.248 | 130 | 2.39 2.72 3.06 / 2.35 2.81 3.02 | 0.73 1.10 1.47 / 0.67 1.08 1.42 |
| C 195E | 2.74, 1.06 | 0.494 | 0.444, 0.217 | 132 | 2.40 2.74 3.07 / 2.33 2.77 3.07 | 0.70 1.06 1.42 / 0.63 1.03 1.43 |
| C 210E | 2.75, 1.14 | 0.495 | 0.416, 0.268 | 140 | 2.39 2.75 3.11 / 2.31 2.81 3.08 | 0.80 1.14 1.48 / 0.77 1.13 1.45 |
| F Block | 2.70, 1.12 | 0.491 | 0.452, 0.195 | 115 | 2.44 2.70 2.96 / 2.40 2.72 2.92 | 0.70 1.12 1.54 / 0.62 1.16 1.51 |
| F 150E | 2.68, 1.42 | 0.346 | 0.281, 0.201 | 118 | 2.45 2.68 2.90 / 2.42 2.68 2.86 | 1.15 1.42 1.68 / 1.15 1.36 1.69 |
| F 165E | 2.71, 1.07 | 0.508 | 0.470, 0.194 | 105 | 2.49 2.71 2.94 / 2.44 2.72 2.90 | 0.61 1.07 1.52 / 0.52 1.09 1.52 |
| F 180E | 2.72, 1.05 | 0.490 | 0.460, 0.168 | 116 | 2.47 2.72 2.96 / 2.39 2.74 2.94 | 0.63 1.05 1.47 / 0.52 1.09 1.49 |
| F 195E | 2.69, 1.15 | 0.472 | 0.435, 0.182 | 123 | 2.41 2.69 2.97 / 2.37 2.73 2.93 | 0.77 1.15 1.53 / 0.68 1.20 1.50 |
| F 210E | 2.67, 1.15 | 0.504 | 0.466, 0.192 | 127 | 2.35 2.67 2.99 / 2.32 2.71 2.95 | 0.76 1.15 1.54 / 0.70 1.17 1.55 |

155

Table S8b. Overlap % of SD2 ellipse fit (NOx vs HOOH) for tropical Pacific (150E-210E, 20S-20N, 0-12 km) and single longitude transects shown in Fig.8.

| | 150E-210E | 150E | 165E | 180E | 195E | 210E |
|---|---|---|---|---|---|---|
| Model C | *100%* | 63% | 87% | 94% | 91% | 89% |
| Models C x F | 60% | | | | | |
| Model F | *100%* | 42% | 88% | 86% | 90% | 86% |

Overlap % is defined as the overlapping area divided by the average of the two ellipse areas.  The Models C x F denotes the overlap % of the two models for the ALL case of 150E-210E.

| Table S5. 1D statistics (mean ± standard deviation) for probability densities shown in Fig 5. | | | | | | | | | | | | | |
|---|---|---|---|---|---|---|---|---|---|---|---|---|---|
| model | | CO | | | | HOOH | | | | $H_2O$ | | | |
| | | Air | PO3 | LO3 | LCH4 | Air | PO3 | LO3 | LCH4 | Air | PO3 | LO3 | LCH4 |
| **A** | -1σ | 4.69 | 4.69 | 4.68 | 4.68 | 2.30 | 2.28 | 2.43 | 2.42 | 2.79 | 2.51 | 3.29 | 3.28 |
| | **avg** | **4.72** | **4.72** | **4.72** | **4.72** | **2.66** | **2.64** | **2.76** | **2.75** | **3.54** | **3.25** | **3.87** | **3.87** |
| | +1σ | 4.76 | 4.76 | 4.76 | 4.76 | 3.01 | 2.99 | 3.09 | 3.09 | 4.30 | 3.98 | 4.44 | 4.46 |
| **B** | -1σ | 4.77 | 4.76 | 4.77 | 4.77 | 2.40 | 2.29 | 2.58 | 2.58 | 2.72 | 2.44 | 3.29 | 3.29 |
| | **avg** | **4.81** | **4.80** | **4.81** | **4.81** | **2.68** | **2.62** | **2.78** | **2.78** | **3.51** | **3.20** | **3.86** | **3.88** |
| | +1σ | 4.84 | 4.83 | 4.85 | 4.85 | 2.97 | 2.94 | 2.98 | 2.99 | 4.30 | 3.97 | 4.44 | 4.47 |
| **C** | -1σ | 4.80 | 4.81 | 4.79 | 4.79 | 2.44 | 2.44 | 2.71 | 2.70 | 2.54 | 2.44 | 3.28 | 3.23 |
| | **avg** | **4.86** | **4.86** | **4.85** | **4.85** | **2.80** | **2.78** | **2.97** | **2.96** | **3.35** | **3.23** | **3.88** | **3.86** |
| | +1σ | 4.92 | 4.92 | 4.90 | 4.90 | 3.15 | 3.12 | 3.22 | 3.22 | 4.16 | 4.02 | 4.48 | 4.49 |
| **D** | -1σ | 4.80 | 4.79 | 4.80 | 4.79 | 2.61 | 2.44 | 2.84 | 2.70 | 2.90 | 2.56 | 3.44 | 3.04 |
| | **avg** | **4.86** | **4.85** | **4.86** | **4.85** | **3.01** | **2.91** | **3.14** | **3.07** | **3.64** | **3.30** | **3.95** | **3.68** |
| | +1σ | 4.93 | 4.91 | 4.92 | 4.91 | 3.41 | 3.37 | 3.45 | 3.44 | 4.38 | 4.03 | 4.46 | 4.33 |
| **E** | -1σ | 4.77 | 4.77 | 4.77 | 4.77 | 2.32 | 2.39 | 2.43 | 2.42 | 2.86 | 2.67 | 3.42 | 3.39 |
| | **avg** | **4.84** | **4.83** | **4.84** | **4.84** | **2.64** | **2.69** | **2.74** | **2.73** | **3.63** | **3.40** | **3.93** | **3.93** |
| | +1σ | 4.90 | 4.90 | 4.91 | 4.91 | 2.96 | 2.99 | 3.05 | 3.05 | 4.39 | 4.13 | 4.44 | 4.46 |
| **F** | -1σ | 4.78 | 4.78 | 4.78 | 4.78 | 2.49 | 2.44 | 2.66 | 2.66 | 2.64 | 2.52 | 3.31 | 3.30 |
| | **avg** | **4.86** | **4.86** | **4.84** | **4.84** | **2.75** | **2.70** | **2.86** | **2.85** | **3.47** | **3.29** | **3.88** | **3.88** |
| | +1σ | 4.93 | 4.95 | 4.90 | 4.89 | 3.01 | 2.97 | 3.05 | 3.05 | 4.29 | 4.06 | 4.45 | 4.47 |

| Table S5 (continued). | | | | | | | | | | | | | |
|---|---|---|---|---|---|---|---|---|---|---|---|---|---|
| model | | HCHO | | | | NOx | | | | $O_3$ | | | |
| | | Air | PO3 | LO3 | LCH4 | Air | PO3 | LO3 | LCH4 | Air | PO3 | LO3 | LCH4 |
| **A** | -1σ | 1.87 | 1.85 | 2.07 | 2.06 | 0.49 | 0.89 | 0.39 | 0.37 | 4.47 | 4.60 | 4.44 | 4.43 |
| | **avg** | **2.17** | **2.14** | **2.32** | **2.31** | **1.05** | **1.36** | **0.90** | **0.89** | **4.70** | **4.80** | **4.66** | **4.65** |
| | +1σ | 2.48 | 2.43 | 2.57 | 2.56 | 1.61 | 1.84 | 1.41 | 1.42 | 4.93 | 4.99 | 4.88 | 4.87 |
| **B** | -1σ | 1.93 | 1.85 | 2.12 | 2.13 | 0.50 | 0.90 | 0.39 | 0.36 | 4.27 | 4.41 | 4.24 | 4.22 |
| | **avg** | **2.20** | **2.13** | **2.33** | **2.34** | **1.03** | **1.32** | **0.88** | **0.86** | **4.51** | **4.62** | **4.47** | **4.45** |
| | +1σ | 2.47 | 2.40 | 2.54 | 2.55 | 1.56 | 1.74 | 1.37 | 1.36 | 4.76 | 4.83 | 4.70 | 4.68 |
| **C** | -1σ | 1.89 | 1.91 | 2.20 | 2.19 | 0.78 | 0.91 | 0.66 | 0.66 | 4.32 | 4.36 | 4.28 | 4.28 |
| | **avg** | **2.22** | **2.22** | **2.43** | **2.43** | **1.14** | **1.27** | **1.00** | **1.01** | **4.49** | **4.54** | **4.45** | **4.45** |
| | +1σ | 2.55 | 2.52 | 2.66 | 2.66 | 1.51 | 1.64 | 1.34 | 1.36 | 4.67 | 4.72 | 4.62 | 4.62 |
| **D** | -1σ | 1.69 | 1.75 | 1.79 | 1.80 | 0.43 | 0.88 | 0.43 | 0.63 | 4.16 | 4.35 | 4.15 | 4.24 |
| | **avg** | **1.89** | **1.97** | **1.97** | **2.00** | **0.98** | **1.36** | **0.89** | **1.12** | **4.40** | **4.56** | **4.38** | **4.47** |
| | +1σ | 2.10 | 2.19 | 2.15 | 2.19 | 1.53 | 1.84 | 1.35 | 1.61 | 4.65 | 4.77 | 4.60 | 4.71 |
| **E** | -1σ | 1.94 | 1.93 | 2.14 | 2.13 | 0.52 | 0.85 | 0.48 | 0.46 | 4.17 | 4.31 | 4.19 | 4.17 |
| | **avg** | **2.21** | **2.19** | **2.34** | **2.34** | **0.97** | **1.22** | **0.92** | **0.90** | **4.41** | **4.53** | **4.42** | **4.41** |
| | +1σ | 2.47 | 2.45 | 2.54 | 2.54 | 1.42 | 1.59 | 1.36 | 1.35 | 4.66 | 4.76 | 4.66 | 4.65 |
| **F** | -1σ | 1.89 | 1.84 | 2.14 | 2.13 | 0.75 | 1.02 | 0.65 | 0.63 | 4.29 | 4.36 | 4.27 | 4.26 |
| | **avg** | **2.21** | **2.15** | **2.37** | **2.37** | **1.17** | **1.37** | **1.03** | **1.02** | **4.49** | **4.54** | **4.46** | **4.44** |
| | +1σ | 2.52 | 2.46 | 2.60 | 2.60 | 1.59 | 1.73 | 1.41 | 1.41 | 4.69 | 4.72 | 4.65 | 4.63 |

160